# Decoding polyubiquitin regulation of K$_V$7.1 (KCNQ1) surface expression with engineered linkage-selective deubiquitinases

Sri Karthika Shanmugam [1,5], Scott A. Kanner [2,5], Xinle Zou[3], Enoch Amarh [1], Papiya Choudhury[1], Rajesh Soni [4], Robert S. Kass[3] & Henry M. Colecraft [1,3] ✉

Polyubiquitin chain diversity generates a 'ubiquitin code' that universally regulates protein abundance, localization, and function. Functions of polyubiquitin diversity are mostly unknown, with lack of progress due to an inability to selectively tune protein polyubiquitin linkages in live cells. We develop linkage-selective engineered deubiquitinases (enDUBs) by fusing linkage-selective DUB catalytic domains to GFP-targeted nanobody and use them to investigate polyubiquitin linkage regulation of an ion channel, YFP-KCNQ1. YFP-KCNQ1 in HEK293 cells has polyubiquitin chains with K48/K63 linkages dominant. EnDUBs yield unique effects on channel surface abundance with a pattern indicating: K11 promotes ER retention/degradation, enhances endocytosis, and reduces recycling; K29/K33 promotes ER retention/degradation; K63 enhances endocytosis and reduces recycling; and K48 is necessary for forward trafficking. EnDUB effects differ in cardiomyocytes and on KCNQ1 disease mutants, emphasizing ubiquitin code mutability. The results reveal distinct polyubiquitin chains control different aspects of KCNQ1 abundance and subcellular localization and introduce linkage-selective enDUBs as potent tools to demystify the polyubiquitin code.

Ubiquitin is a pervasive regulator of protein expression and cell signaling pathways and, thereby, controls all aspects of biology. Protein ubiquitination is accomplished through the sequential activities of three enzyme classes– E1 enzymes catalyze ATP-dependent activation and transfer of ubiquitin to E2 conjugating enzymes from where it is covalently attached most frequently to the ε-amino group of substrate lysine residues through the activity of E3 ubiquitin ligases[1,2]. The human genome encodes 2 E1, ~40 E2, and >600 E3 enzymes. Ubiquitin itself has seven internal lysine residues (K6, K11, K27, K29, K33, K48, and K63) which, along with its N-terminal M1 residue, can be ubiquitinated to produce a variety of polyubiquitin chains with distinctive structures[3,4]. The capacity to produce specific polyubiquitin linkages is encoded within particular E2 and E3 ligases. Individual proteins may be modified by monoubiquitination, multi-monoubiquitination, homotypic polyubiquitination (one linkage type), and/or heterotypic polyubiquitination with mixed and/or branched chains[1,3,5,6]. The distinctive topological features of diverse ubiquitin/polyubiquitin modifications are decoded by various effector proteins with domains and motifs that recognize specific ubiquitin configurations to yield discrete functional outcomes[7–9]. Protein ubiquitination is dynamically countered by ~100 deubiquitinases (DUBs) which hydrolyze ubiquitin chains in distinctive ways[10]. Altogether, the system of ubiquitin writers (E1, E2 and E3 ligases), readers (effectors), editors, and erasers (DUBs), together with the multiplicity of possible ubiquitin modification types on proteins, constitute a complex and versatile 'ubiquitin code' that remains largely enigmatic despite decades of research[1,5,6,11]. The scope of the problem

[1]Department of Physiology and Cellular Biophysics, Columbia University Irving Medical Center, New York, NY, USA. [2]Doctoral Program in Neurobiology and Behavior, Columbia University Irving Medical Center, New York, NY, USA. [3]Department of Molecular Pharmacology and Therapeutics, Columbia University Irving Medical Center, New York, NY, USA. [4]Proteomics and Macromolecular Crystallography, Columbia University Irving Medical Center, New York, NY, USA. [5]These authors contributed equally: Sri Karthika Shanmugam, Scott A. Kanner. ✉e-mail: hc2405@cumc.columbia.edu

is colossal due to the prevalence of ubiquitination, the uniqueness of distinct cellular contexts, and the likely substrate-specific nature of functional outcomes. A generalizable method that permits substrate-selective, linkage-selective hydrolysis of polyubiquitin chains from target proteins in live cells is necessary for deciphering the mysterious ubiquitin code, but is currently lacking.

Cell surface ion channels are a specialized class of integral membrane proteins that are present in all organs, tissues, and cell types and are necessary for myriad biological processes[12]. The precise mechanisms regulating their stability and trafficking to and away from the cell surface through various intracellular compartments are largely mysterious, but known to be influenced by ubiquitination[13,14]. Many devastating ion channelopathies arise from mutations in ion channels that adversely impact their surface density, highlighting the importance of understanding fundamental ion channel trafficking mechanisms[15,16]. Linkage-selective polyubiquitination is a potentially critical, but under-appreciated determinant of ion channel trafficking among distinct subcellular compartments. These general concepts are exemplified by KCNQ1 (Kv7.1) which together with auxiliary KCNE1 subunits generates the slow delayed rectifier current, $I_{Ks}$, necessary for proper repolarization of human cardiac ventricular action potential[17,18]. KCNQ1 also underlies $K^+$ currents in the inner ear. Loss-of-function mutations in KCNQ1 cause long QT syndrome type 1 (LQT1), exercise-induced sudden cardiac death, and deafness[19–22]. While it is clear that KCNQ1 stability and surface density are down-regulated by ubiquitination[23,24], the role of distinct polyubiquitin linkage types in these processes are unknown.

In this work, we develop a suite of engineered deubiquitinases (enDUBs) generated by fusing an anti-GFP/YFP nanobody to catalytic domains of DUBs with distinctive polyubiquitin chain preferences, and apply these to investigate the roles of divergent polyubiquitin chains on KCNQ1-YFP abundance, subcellular localization, and function. The results reveal a prominent role for distinct polyubiquitin chains in regulating KCNQ1 trafficking among separate subcellular compartments, demonstrate the mutability of the ubiquitin code for a substrate depending on both extrinsic (cell context) and intrinsic (mutations) factors, and introduce the utility of linkage-selective enDUBs as a powerful method to unravel multifaceted roles of the polyubiquitin code in live cells.

## Results

### KCNQ1-YFP subcellular distribution and polyubiquitination in HEK293 cells

We used confocal microscopy and immunofluorescence to visualize steady-state distribution of KCNQ1-YFP in subcellular compartments—endoplasmic reticulum (ER), Golgi, early endosomes (EE), late endosomes (LE), lysosomes and proteasomes—after transient transfection in HEK293 cells. Most of the channel fluorescence co-localized with the ER, but we also observed significant co-localization with all the other subcellular markers, indicating robust steady-state distribution in these compartments (Fig. 1A, B). To selectively visualize and quantify plasma membrane channels, we engineered a 13-residue high-affinity bungarotoxin binding site (BBS) into an extracellular loop of YFP-tagged KCNQ1 (BBS-KCNQ1-YFP) and incubated non-permeabilized transfected cells with Alexafluor-647-conjugated α-bungarotoxin (BTX-647) (Fig. 1C)[24,25]. We used flow cytometry to rapidly measure YFP and BTX-647 fluorescence as metrics of total KCNQ1 expression and channel surface density, respectively (Fig. 1D). Transfected cells treated with MG132 to inhibit the proteasome moderately increased total and surface KCNQ1-YFP (Fig. 1D), enhanced channel co-localization with the ER and Golgi, and decreased association with EE and lysosomal compartments (Supplemental Fig. 1). Notably, MG132 caused visually striking changes in the appearance of the ER (more clumped presentation) and lysosomes (Supplemental Fig. 1). By contrast with MG132, treating cells with chloroquine to eliminate

lysosomal degradation yielded minimal effects on KCNQ1-YFP abundance and surface density (Supplemental Fig. 2), moderately enhanced ER association and decreased colocalization with LE and proteasomal compartments (Supplemental Fig. 1). Thus, KCNQ1-YFP transiently expressed in HEK293 cells is differentially distributed among distinct subcellular compartments and degraded predominantly via the proteasome.

The precise molecular signals underlying KCNQ1 subcellular localization and degradation are unknown. We hypothesized that distinct polyubiquitin linkage chains could represent a prominent mechanism for controlling KCNQ1-YFP abundance and subcellular localization. We used mass spectrometry to determine the prevalence of polyubiquitin chains on KCNQ1-YFP expressed in HEK293 cells. KCNQ1-YFP was pulled down in transiently transfected cells using anti-KCNQ1 antibody and detected by Coomassie staining following gel electrophoresis (Fig. 1E). Bands corresponding to KCNQ1-YFP monomer and dimers were excised for mass spectrometry analysis, using corresponding extracted segments from non-transfected cells as control (Fig. 1E). In addition to the expected KCNQ1 peptides, we identified enrichment of ubiquitin peptides compared to control, a fraction of which had di-glycines on lysine residues, a tell-tale signature of ubiquitin modification. Within the population of di-glycine modified ubiquitin lysine residues, K48 was fractionally dominant (72%) followed by K63 (24%), while other atypical chains (K11, K27, K29, K33, and K6) were much less prevalent (4%) (Fig. 1F, G; and Supplemental Fig. 3).

### Rationale and design of linkage-selective and non-specific enDUBs

While the mass spectrometry results suggest the presence of distinct polyubiquitin linkages on KCNQ1-YFP expressed in HEK293 cells, this data does not provide any information on whether and how distinct polyubiquitin chain linkages regulate KCNQ1 abundance, subcellular localization, and function. We sought to develop an approach to systematically examine the potential role of distinct polyubiquitin linkages in regulating KCNQ1 abundance and localization in live cells by exploiting minimal catalytic domains of deubiquitinases (DUBs) with preferences for hydrolyzing particular polyubiquitin linkages: OTUD1 (O1)−K63; OTUD4 (O4) - K48; Cezanne (Cz)−K11; TRABID (Tr) - K29/K33; and USP21 (U21)−non-specific[10,26,27]. We fused the catalytic subunits of these DUBs to a GFP-targeted nanobody[28] to generate putative linkage-selective engineered DUBs (enDUBs) that would act selectively on GFP/YFP-tagged protein substrates (Fig. 1H; and Supplemental Fig. 4)[29]. We tested how effectively the different enDUBs decreased basal ubiquitination of KCNQ1-YFP. In control HEK293 cells expressing KCNQ1-YFP and the unmodified GFP nanobody (nano), immunoprecipitation followed by immunoblotting with anti-KCNQ1 yielded bands representing KCNQ1 monomers, dimers, trimers and tetramers (Fig. 1I, top). Stripping this blot and probing with anti-ubiquitin showed a smear consistent with basal KCNQ1 ubiquitination (Fig. 1I, bottom), and in accordance with the mass spectrometry results (Fig. 1F, G). Co-expressing KCNQ1-YFP with the different enDUBs all resulted in significant decreases in global channel ubiquitination, albeit with quantitative differences in efficacy (Fig. 1I, J). EnDUBs were expressed in a P2A-CFP plasmid, permitting detection of their expression by monitoring CFP fluorescence and by Western blot (Supplemental Fig. 5). In control experiments, catalytically dead enDUBs did not decrease KCNQ1-YFP ubiquitination, and in some cases appeared to increase ubiquitination, potentially reflecting a shielding effect (Supplemental Fig. 6).

### Differential impact of distinct enDUBs on KCNQ1 expression, subcellular localization, and function

The relatively uniform efficacy of all five enDUBs in reducing the amount of ubiquitin on pulled down KCNQ1-YFP (as reported by a

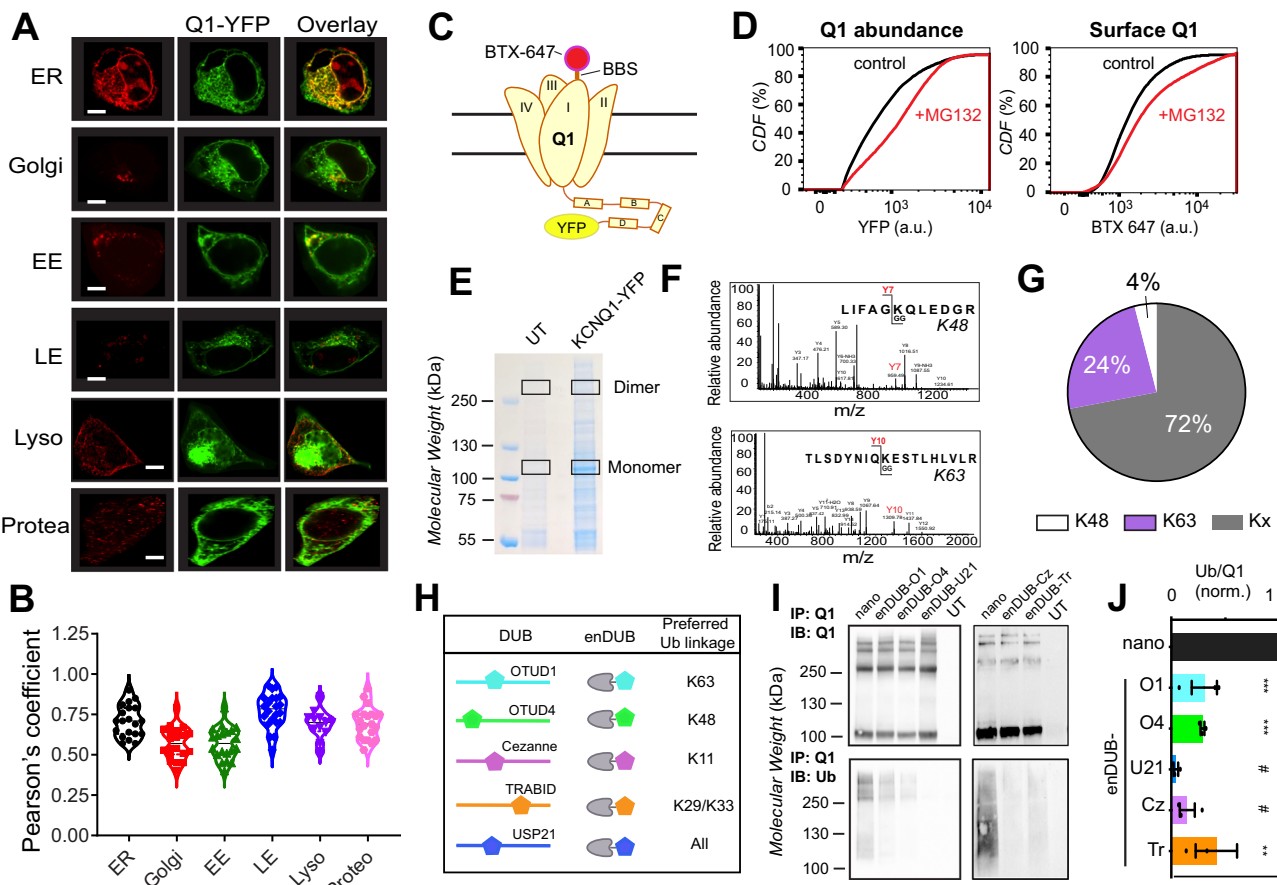

**Fig. 1 | KCNQ1-YFP subcellular distribution and polyubiquitination in HEK293 cells and design of linkage-selective enDUBs. A** Representative confocal images of HEK293 cells showing co-localization of expressed KCNQ1-YFP (green) and immunolabelled subcellular organelle markers (red) [calnexin - endoplasmic reticulum (ER); RCAS1 – Golgi; EEA1 - early endosome (EE); Rab9A - late endosome (LE); LAMP2 - lysosome (lyso); and PSMA2 - proteasome (protea)]. Scale bar, 5 µm. **B** Co-localization of Q1-YFP with subcellular markers assessed by Pearson's co-localization coefficient. ($N$ = 3 independent experiments; $n$ > 20 cells for each subcellular organelle imaged. **C** Schematic of BBS-KCNQ1-YFP with BTX-647 bound to an extracellular bungarotoxin-binding site (BBS). **D** Representative flow cytometry cumulative distribution function (CDF) plots showing total (YFP fluorescence) and surface (BTX-647) channels in YFP-positive cells expressing BBS-KCNQ1-YFP in untreated (black) and MG132 treated (red) conditions. Here and throughout, appropriate gates were used to determine cells vs debris (FSC/SSC) and single cells vs doublets (FSC-A/FSC-H). Fluorescence gates (CFP, YFP, BTX-647) were determined from single color controls compared to untransfected cells. **E** Representative Coomassie-stained SDS PAGE of pulled down KCNQ1-YFP. Black boxes indicate KCNQ1 monomeric and dimeric protein bands excised for mass spectrometry. **F** Representative ms2 spectra traces for ubiquitin peptides with di-glycine modification of K48 (*top*) and K63 (*bottom*), respectively. **G** Fractional distribution of Ub linkages on pulled down KCNQ1-YFP as assessed by mass spectrometry of associated di-glycine modified lysine residues on associated ubiquitin peptides. **H** Schematic of linkage-selective engineered deubiquitinases (enDUBs). **I** *Top*, KCNQ1-YFP pulldowns probed with anti-KCNQ1 in untransfected cells (UT), cells expressing KCNQ1-YFP with nano (control) or the indicated enDUBs. The four bands represent KCNQ1-YFP monomeric, dimeric, trimeric and tetrameric species. *Bottom*, same blots stripped and probed with anti-ubiquitin. **J** Relative KCNQ1-YFP ubiquitination computed as the ratio of ubiquitin/(monomeric + dimeric KCNQ1) signal intensities here and throughout (mean ± SEM); $N$ = 3 biological replicates. One-way ANOVA and Dunnett's multiple comparisons test, #$p$ < 0.0001, *** $p$ < 0.0003 and **$p$ = 0.0016. Source data are provided as a Source Data file.

pan-ubiquitin antibody) was somewhat surprising given the mass spectrometry data showing asymmetry in linkage types (Fig. 1G). There were three possibilities to explain this apparent discrepancy. First, in cells, the over-expressed enDUBs indiscriminately hydrolyzed all ubiquitin linkages in a substrate-independent manner. Second, prolonged tethering of the enDUBs to YFP enabled by the nanobody resulted in hydrolysis of all ubiquitin chain types present specifically on KCNQ1-YFP[26,27]. Third, there is a prevalence of branched polyubiquitin chains with mixed linkages such that hydrolysis of even minor linkage species (atypical chains, such as K11, K29, K33, K6) can result in removal of a significant fraction of total ubiquitin associated with the channel. In the case of the first two possibilities, the distinct enDUBs would be expected to have a similar functional impact on channel abundance, localization, and function, whereas divergent effects could be possible in the third scenario. We assessed the impact of the different enDUBs

on KCNQ1-YFP abundance, localization, and function to distinguish among these possibilities.

We first used flow cytometry to contrast the impact of different enDUBs on BBS-KCNQ1-YFP total expression and surface density (Fig. 2A–E). Putatively targeting K63 hydrolysis with enDUB-O1 had no impact on either BBS-KCNQ1-YFP total expression or surface density. EnDUB-O4 (K48) also had no effect on KCNQ1 total expression, but unexpectedly reduced channel surface density. A different enDUB also targeting K48 linkages (enDUB-OTUB1) also decreased KCNQ1 surface expression similar to enDUB-O4 (Supplemental Fig. 7). By contrast, enDUB-Cz (K11) and enDUB-Tr (K29/K33) both significantly increased KCNQ1 abundance and surface density (Fig. 2B–E). Finally, enDUB-U21, which hydrolyzes all chain types, increased KCNQ1 total expression without impacting channel surface density. In control experiments, none of the enDUBs affected BBS-KCNQ1 surface density when no YFP

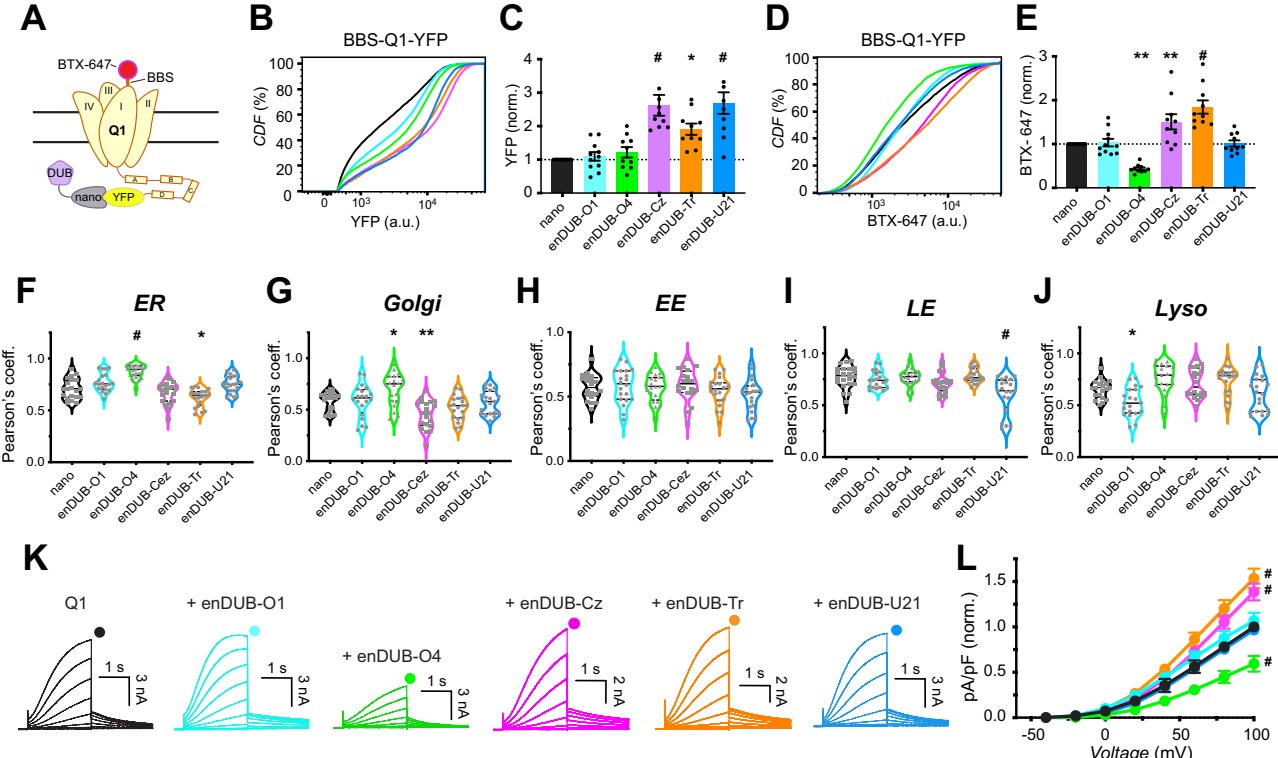

**Fig. 2 | Differential impact of distinct enDUBs on KCNQ1 steady-state abundance, subcellular localization, and function.** **A** Experimental design schematic; BBS-KCNQ1-YFP is co-expressed with either GFP/YFP targeting nanobody (nano, control) or an enDUB in HEK293 cells. **B** Representative flow cytometry CDF plots showing channel abundance (YFP fluorescence) in cells expressing BBS-Q1-YFP and either nano alone (black) or each of the enDUBs (enDUB-O1, cyan; enDUB-O4, green; enDUB-Cz, magenta; enDUB-Tr, orange; enDUB-U21, blue). **C** Quantification of flow cytometry experiments for total KCNQ1-YFP expression analyzed from YFP- and CFP- positive cells ($N = 10$ biological replicates; #$p < 0.0001$, one-way ANOVA with Dunnett's multiple comparisons: $p = 0.9970$, nano vs enDUB-O1; $p = 0.9136$, nano vs enDUB-O4; #$p < 0.0001$, nano vs enDUB-Cz; *$p = 0.0117$, nano vs enDUB-Tr; #$p < 0.0001$, nano vs enDUB-U21). Data were normalized to values from the nano control group (dotted line) and presented as mean ± SEM. **D** Representative CDF plots showing surface fluorescence (BTX-647) in cells expressing BBS-KCNQ1-YFP and either nano alone (black) or an enDUB (enDUB-O1, cyan; enDUB-O4, green; enDUB-Cz, magenta; enDUB-Tr, orange; enDUB-U21, blue). **E** Quantification of flow cytometry experiments for KCNQ1-YFP surface expression analyzed from YFP- and CFP- positive cells ($N = 10$ biological replicates; #$p < 0.0001$, one-way ANOVA with Dunnett's multiple comparisons; $p = 0.9995$, nano vs enDUB-O1; **$p = 0.0012$, nano vs enDUB-O4; **$p = 0.0041$, nano vs enDUB-Cz; #$p < 0.0001$, nano vs enDUB-Tr; $p > 0.9999$, nano vs enDUB-U21). Data were normalized to values from the nano control group (dotted line) and presented as mean ± SEM. **F**–**J** Co-localization of KCNQ1-YFP with subcellular markers assessed by Pearson's co-localization coefficient ($N = 3$, $n > 20$ for each subcellular organelle; ER #$p < 0.0001$, one-way ANOVA

and Dunnett's multiple comparisons test: $p = 0.01125$, nano vs enDUB-O1; #$p < 0.0001$, nano vs enDUB-O4; $p = 0.4620$, nano vs enDUB-Cz; *$p = 0.0109$, nano vs enDUB-Tr; $p = 0.3397$, nano vs enDUB-U21. Golgi #$p < 0.0001$, one-way ANOVA and Dunnett's multiple comparisons test: $p = 0.9999$, nano vs enDUB-O1; *$p = 0.0213$, nano vs enDUB-O4; **$p = 0.0080$, nano vs enDUB-Cz; $p = 0.4119$, nano vs enDUB-Tr; $p = 0.9990$, nano vs enDUB-U21. EE $p = 0.3250$, one-way ANOVA and Dunnett's multiple comparisons test: $p = 0.9992$, nano vs enDUB-O1; $p = 0.9998$, nano vs enDUB-O4; $p = 0.9845$, nano vs enDUB-Cz; $p = 0.8955$, nano vs enDUB-Tr; $p = 0.3685$, nano vs enDUB-U21. LE #$p < 0.0001$, one-way ANOVA and Dunnett's multiple comparisons test: $p = 0.8850$, nano vs enDUB-O1; $p > 0.9999$, nano vs enDUB-O4; $p = 0.2278$, nano vs enDUB-Cz; $p = 0.9927$, nano vs enDUB-Tr; #$p < 0.0001$, nano vs enDUB-U21. Lyso #$p < 0.0001$, one-way ANOVA and Dunnett's multiple comparisons test: *$p = 0.0209$, nano vs enDUB-O1; $p = 0.0645$, nano vs enDUB-O4; $p = 0.4973$, nano vs enDUB-Cz; $p = 0.1985$, nano vs enDUB-Tr; $p = 0.7449$, nano vs enDUB-U21. **K** Exemplar KCNQ1 + KCNE1 current traces from whole-cell patch clamp measurements in CHO cells. **L** Population *I-V* curves for nano control (black, $n = 25$), enDUB-O1 (cyan, $n = 9$), enDUB-O4 (green, $n = 9$), enDUB-Cz (magenta, $n = 16$), enDUB-Tr (orange, $n = 16$) and enDUB-U21 (blue, $n = 8$). Pooled data were collected across $N = 3$ independent experiment days and presented as mean ± SEM; #$p < 0.0001$, two-way ANOVA, with Tukey's multiple comparisons: $p = 0.1996$, nano vs enDUB-O1; #$p < 0.0001$, nano vs enDUB-O4; #$p < 0.0001$, nano vs enDUB-Cz; #$p < 0.0001$, nano vs enDUB-Tr; $p = 9463$, nano vs enDUB-U21. Source data is provided as a Source Data file.

was attached to the channel, indicating the necessity of enDUB targeting to the substrate to yield a functional effect (Supplemental Fig. 8). The distinct enDUBs also produced characteristic changes to KCNQ1-YFP intracellular organelle distribution (Fig. 2F–J; and Supplemental Fig. 9). By comparison to control (nano), enDUB-Tr and enDUB-Cz decreased co-localization with the ER and Golgi, respectively, whereas enDUB-O4 increased both ER and Golgi retention (Fig. 2F–G); enDUB-U21 decreased colocalization with LE (Fig. 2I); and enDUB-O1 diminished association with the lysosome (Fig. 2J).

We next employed whole-cell patch clamp electrophysiology to determine the functional impact of the different enDUBs on KCNQ1 ionic currents (Fig. 2K, L). Control cells expressing KCNQ1-YFP + KCNE1 + nano displayed characteristic robust slowly activating delayed

rectifier K⁺ currents ($I_{Ks}$) (Fig. 2K)[18,25]. In concordance with the observed effects on channel surface density, enDUB-O1 and enDUB-U21 had minimal effects on $I_{Ks}$, enDUB-O4 significantly decreased $I_{Ks}$, while enDUB-Cz and enDUB-Tr both increased $I_{Ks}$ (Fig. 2K, L).

Overall, these results revealed a number of unexpected findings including that: hydrolysis of apparently minor polyubiquitin linkage chains species (K11 and K29/K33) yielded the most robust stabilization of KCNQ1 expression; hydrolysis of different polyubiquitin linkages distinctively alter KCNQ1 distribution among various intracellular compartments; that hydrolysis of distinct ubiquitin chains can selectively suppress or upregulate KCNQ1 surface density independent of effects on channel abundance; and that K48 polyubiquitin, the well-established canonical degradation signal, does not control KCNQ1

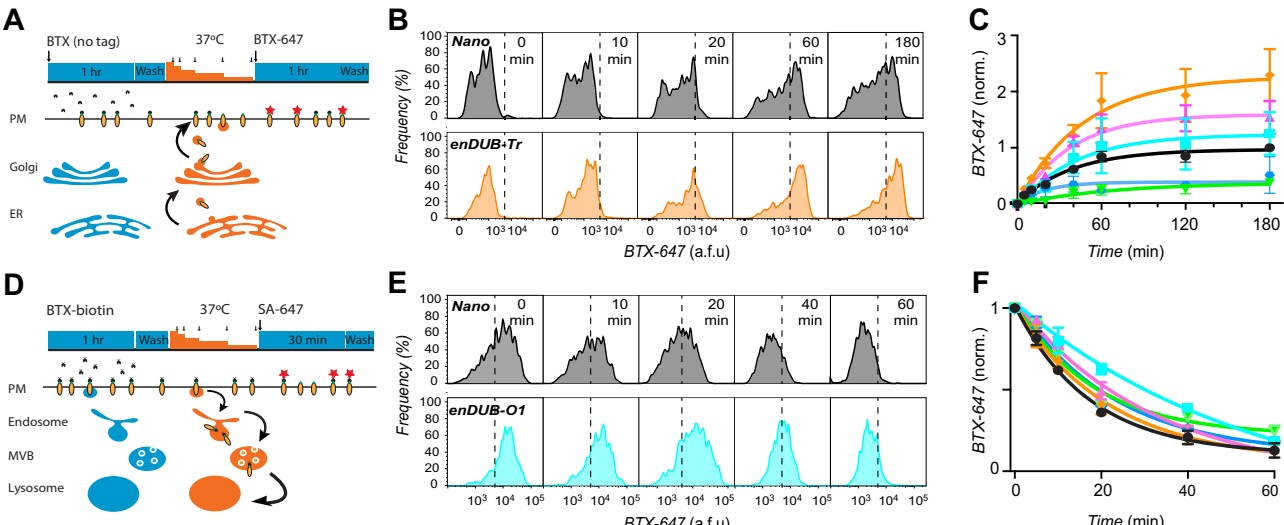

**Fig. 3 | Distinctive impact of different enDUBs on dynamic KCNQ1 delivery to and removal from the plasma membrane. A** Schematic of optical pulse-chase assay to measure BBS-KCNQ1-YFP forward trafficking. **B** Representative flow cytometry BTX-647 fluorescence histograms showing time evolution of surface channel increase in cells expressing BBS-KCNQ1-YFP with nano (black, *top*) or enDUB-Tr (orange, *bottom*). **C** Time evolution of channel forward trafficking to the surface in cells expressing BBS-KCNQ1-YFP with either nano or an enDUB (enDUB-O1, cyan; enDUB-O4, green; enDUB-Cez, magenta; enDUB-Tr, orange; enDUB-U21, blue; $N = 3$ for each data point; $p = 0.8047$, nano vs enDUB-O1; $p < 0.0001$, nano vs enDUB-O4; $p = 0.0014$, nano vs enDUB-Cz; $p < 0.0001$, nano vs enDUB-Tr; $p = 0.0053$, nano vs enDUB-U21; two-way ANOVA, with Tukey's multiple comparisons). Data are normalized to the max value of nano control (black) and presented as mean ± SEM. Smooth curves are fits of an exponential growth function to the data: $y = Ae^{1/\tau} + y0$. **D** Schematic of endocytosis pulse-chase assay. **E** Representative flow cytometry BTX-647 fluorescence histograms showing time evolution of surface channel decrease in cells expressing BBS-KCNQ1-YFP with nano (black, *top*) or enDUB-O1 (turquoise, bottom). **F** Time evolution of channel removal from the surface in cells expressing BBS-KCNQ1-YFP with either nano ($t_{1/2} = 12.02$ mins or an enDUB (enDUB-O1, cyan, $t_{1/2} = 38.32$ mins; enDUB-O4, green, $t_{1/2} = 12.43$ mins; enDUB-Cz, magenta, $t_{1/2} = 21.79$ mins; enDUB-Tr, orange, $t_{1/2} = 15.06$ mins; enDUB-U21, blue, $t_{1/2} = 15.61$ mins; $N = 3$ for each data point; $p < 0.0001$, nano vs enDUB-O1, $p = 0.0171$, nano vs enDUB-O4, $p = 0.0104$, nano vs enDUB-Cz, $p = 0.9326$, nano vs enDUB-Tr, $p = 0.0708$, nano vs enDUB-U21, two-way ANOVA with Tukey's multiple comparisons). Data are presented as mean values ± SEM. Smooth curves are fits of an exponential growth function to the data: $y = Ae^{-1/\tau} + y0$. Source data is provided as a Source Data file.

abundance in this context but rather is important for channel trafficking to the cell surface. The results also validate the linkage-selective enDUB strategy as capable of providing unique insights into the in situ functions of distinct polyubiquitin linkage types in a substrate-specific manner.

## Distinctive impact of enDUBs on KCNQ1 delivery to and removal from the plasma membrane

The steady-state surface density of KCNQ1 is maintained by dynamic processes of channel delivery to and removal from the plasma membrane. KCNQ1 delivery to the surface involves both the transport of newly synthesized channels via the Golgi (or directly from the ER) and the recycling of a fraction of endocytosed channels. We wondered whether the divergent effects of the enDUBs on KCNQ1 steady-state surface density were mediated by differential regulation of dynamic forward trafficking and internalization processes. To address this question, we utilized optical-pulse chase assays to determine rates of BBS-KCNQ1-YFP delivery to and removal from the plasma membrane and assessed the impact of the enDUBs on these processes (Fig. 3).

To monitor channel delivery to the plasma membrane, HEK293 cells transiently expressing BBS-Q1-YFP were initially incubated at 4 °C to arrest trafficking processes after which the non-permeabilized cells were exposed to excess unconjugated (non-fluorescent) BTX to saturate extracellular BBS epitopes (pulse). Cells were then incubated at 37 °C for varying times to resume trafficking (chase), and the newly delivered BBS-Q1-YFP channels labeled with BTX-647 at 4 °C, followed by quantifying fluorescence signals using flow cytometry (Fig. 3A)[24]. In control cells expressing BBS-Q1-YFP, the BTX-647 increased exponentially with a half-time of 20.37 mins, and reaching a plateau after 2 hrs (Fig. 3B, C). To enable facile comparison among different experiments, we normalized BTX-647 signals to the maximum value obtained from the contemporaneous control group (at 3 hrs), following

background subtraction (Fig. 3C). The dominant effects of the distinct enDUBs lay in their dramatically different effects on the amount of BBS-KCNQ1-YFP delivered to the cell surface, as reported by relative plateau levels of the forward trafficking curves (Fig. 3C). Whereas enDUB-O1 had only a marginal effect, enDUB-O4 and enDUB-U21 both led to a marked ~70% decrease in the BTX-647 steady-state signal compared to control, reflecting a strong suppression of channel forward trafficking. In sharp contrast, enDUB-Cz and enDUB-Tr both robustly enhanced KCNQ1 forward trafficking as revealed by 1.6 and 2-fold increases, respectively, in BTX-647 plateau signal relative to control (Fig. 3C).

To measure KCNQ1 endocytosis, BBS-KCNQ1-YFP channels initially at the cell surface were labeled with biotinylated bungarotoxin (BTX-biotin) at 4 °C (pulse). Cells were then incubated at 37 °C for varying time periods to resume trafficking (chase), followed by labeling with streptavidin-conjugated Alexa Fluor 647 (SA-647) at 4 °C (Fig. 3D)[24]. In this paradigm, only channels that were present at the cell surface during the pulse phase and labeled with BTX-biotin would be fluorescently labeled with SA-647 after the chase, distinguishing them from new channels that are subsequently inserted. A decrease in SA-647 fluorescence signal with increasing chase times would be expected due to internalization of BTX-biotin-labeled channels. Indeed, control cells expressing BBS-KCNQ1-YFP + nano displayed an exponential decline in SA-647 fluorescence with chase time, with a half-life ($t_{1/2}$) of 12.02 mins (Fig. 3E, F). The different enDUBs produced distinctive influences on the apparent rate of BBS-KCNQ1-YFP internalization (Fig. 3F). The most dramatic effect was observed with enDUB-O1 which slowed the apparent rate of internalization by over 3-fold ($t_{1/2} = 38.32$ mins) (Fig. 3E, F). EnDUB-Cz caused an intermediate slowing of the apparent rate of internalization ($t_{1/2} = 21.79$ mins), whereas enDUB-O4 ($t_{1/2} = 12.43$ mins), enDUB-Tr ($t_{1/2} = 15.06$ mins), and enDUB-U21 ($t_{1/2} = 15.61$ mins) had marginal effects (Fig. 3F).

Recycling of KCNQ1-YFP is another layer of subcellular dynamics that may be modulated by ubiquitin and may potentially contribute to the measurements of forward trafficking and endocytosis. To determine the potential contribution of channel recycling to BBS-KCNQ1-YFP steady-state surface density, we utilized dominant negative Rab11 (Rab11-DN) to block slow recycling endosomes (Supplemental Fig. 10). We observed a ~20% decrease in BBS-KCNQ1-YFP steady-state density in cells co-expressing Rab11-DN compared to nano (control) (Supplemental Fig. 8). In the presence of Rab11-DN, enDUB-O1 yielded a 50% suppression in BBS-KCNQ1-YFP surface density (Supplemental Fig 10). Co-expressing Rab11-DN also largely neutralized the increase in surface density observed with enDUB-Cz (Supplemental Fig. 10). These data indicate that under baseline conditions (i.e., without Rab11-DN) enDUB-O1 and enDUB-Cz increase the pool of channels recycled to the cell surface after endocytosis. Thus, by inference, K63 and K11 reduce the pool of channels recycled to the surface. By contrast, Rab11-DN did not substantively diminish the increase in BBS-KCNQ1-YFP observed with enDUB-Tr (Fig. 2D, E; and Supplemental Fig. 10) suggesting a minimal impact of K29/K33 on channel recycling. Altogether, these data suggest that pruning K11 enhances forward trafficking from ER/Golgi, slows endocytosis, and promotes recycling; removing K29/K33 from the channel solely enhances forward trafficking from the ER/Golgi; whereas stripping K63 slows endocytosis and increases channel recycling.

## KCNQ1 is downregulated by NEDD4-2 and ITCH with distinct polyubiquitin signatures

The system of writers, erasers, readers, and editors that constitute the ubiquitin code undoubtedly varies among different cell types due to expected differences in protein expression profiles. As such, the ubiquitin code regulation of a particular protein may not be immutable but rather change in different cellular environments. To explore this anticipated but under-explored dimension of the ubiquitin code, and to further test the discriminative capacity of linkage-selective enDUBs, we sought to perturb the polyubiquitin signature of KCNQ1 in HEK293 cells. To this end, the HECT E3 ligase NEDD4-2 which predominantly catalyzes the synthesis of K63 polyubiquitin chains on substrates, is a known regulator of KCNQ1[23,30]. Accordingly, NEDD4-2 co-expression dramatically reduced both KCNQ1 surface density (Fig. 4A, B) and whole-cell currents (Fig. 4C, D). We screened other members of the NEDD4 family of HECT E3 ligases and identified ITCH as a regulator of KCNQ1. Similar to NEDD4-2, ITCH significantly down-regulated both KCNQ1 channel surface density and ionic currents (Fig. 4A–D). The functional effects of NEDD4-2 were diminished by a mutation within a canonical PY motif located on the KCNQ1 C-terminus (Y662A) (Supplemental Fig. 11) as previously reported[23]. Surprisingly, the functional effects of ITCH on KCNQ1 were unaffected by the Y662A mutation (Supplemental Fig. 11), indicating a potential difference in how it associates with the channel compared to NEDD4-2.

We used mass spectrometry to determine the impact of NEDD4-2 and ITCH co-expression, respectively, on KCNQ1 polyubiquitin signatures. Mass spectrometry analyses on KCNQ1 pulldown bands excised from a Coomassie stained gel indicated that NEDD4-2 and ITCH co-expression, respectively, yielded 4.5-fold and 3.5-fold increases in total ubiquitin associated with the channel compared to control (nano) (Fig. 4E). Compared to control, NEDD4-2 increased the relative amount of ubiquitin peptides with di-glycine modifications on K63 (13-fold), K48 (5-fold), and other linkages, Kx (3-fold) (Fig. 4F). ITCH also increased di-glycine modifications of ubiquitin K63 (6-fold), K48 (4-fold), and Kx (4-fold) (Fig. 4F). Further, in the ubiquitin pool pulled down with the channel, NEDD4-2 significantly increased the fraction of K63 ubiquitin linkages (from 24% to 35%) and decreased K48 (from 72% to 60%) while having a modest effect on the fraction of other linkage types (from 4% to 5%) (Fig. 4G) relative to control (Fig. 1G). By contrast, ITCH significantly increased the fraction of atypical linkages Kx (from

4% to 13%), marginally altered the fraction of K63 (from 24% to 26%), and reduced K48 (from 72% to 61%) (Fig. 4G) relative to control (Fig. 1G).

Beyond variations in polyubiquitin linkage types, another potential factor influencing the functional impact of NEDD4-2 and ITCH on KCNQ1-YFP surface expression is the complement of lysine residues on the channel that are ubiquitinated under the different conditions. There are 31 intracellular lysine residues on KCNQ1 that could potentially act as sites for ubiquitin attachment of which 25 are present in the C-terminus (Supplemental Fig. 12). In control cells expressing KCNQ1-YFP + nano, mass spectrometry indicated three lysine residues−K467, K579, and K598−in which the fraction of di-glycine modified peptide exceeded 1% (Fig. 4H, I). NEDD4-2 did not appreciably change the fraction of di-glycine modified K467, K579, and K598, but did result in the emergence of di-glycine modified K581 (Fig. 4H, I). By contrast, ITCH yielded detection of more di-glycine modified lysines on KCNQ1 including K393, K411/K412/K413, K427 and K569 in addition to K467, K579, K581 and K598 seen with NEDD4-2 (Fig. 4H, I).

Overall, these data suggest that the similar functional effects of NEDD4-2 and ITCH on KCNQ1 currents and surface density may be mediated through divergent polyubiquitin signatures. Thus, co-expression of NEDD4-2 and ITCH constitute unique adjustments of the ubiquitin code regulating KCNQ1 surface expression, providing a forum to further test the discriminatory capacity of linkage-selective enDUBs.

## Differential impact of enDUBs on reversing KCNQ1 functional downregulation by NEDD4-2 and ITCH

We first investigated the capacity of the different enDUBs to counter the impact of NEDD4-2 on KCNQ1 ubiquitination and functional expression (Fig. 5). In control cells co-expressing KCNQ1-YFP + NEDD4-2 + nano, pulldown of the channel followed by immunoblotting with KCNQ1 antibodies yielded specific bands consistent with full-length monomeric KCNQ1-YFP and oligomeric species (Fig. 5B, left). Stripping the blot and probing with ubiquitin antibodies yielded strong staining (Fig. 5B, right), consistent with the ~4.5-fold increase in KCNQ1-YFP ubiquitination in the presence of NEDD4-2 overexpression observed by mass spectrometry (Fig. 4E). All five enDUBs when coexpressed with KCNQ1-YFP + NEDD4-2 dramatically reduced the ubiquitin signal on pulled-down KCNQ1-YFP (Fig. 5B,C). Despite the apparently similar impact of the distinct enDUBs on the broad biochemical signature of ubiquitin pulled down with the channel, analyses of their effects on channel abundance, subcellular localization, and ionic currents delineated sharp differences amongst them. KCNQ1 abundance was strongly enhanced by enDUB-Cz/enDUB-U21, more moderately increased by enDUB-O1/enDUB-Tr, and decreased by enDUB-O4 (Fig. 5D, E). The decrease in KCNQ1 surface density caused by NEDD4-2 overexpression was reversed by enDUB-O1 and enDUB-Cz to beyond the level observed under control baseline conditions (Fig. 5F, G). Channel surface density was also rescued by enDUB-U21, although to a smaller extent than observed with enDUB-O1/enDUB-Cz. Surprisingly, enDUB-Tr was completely ineffective in rescuing surface density of KCNQ1-YFP channels down-regulated by NEDD4-2 overexpression (Fig. 5F, G). Whole-cell patch clamp studies further consolidated these results. Consistent with the pattern of rescue of KCNQ1 surface density, $I_{Ks}$ currents suppressed by NEDD4-2 overexpression were strongly rescued by enDUB-O1/enDUB-Cz/enDUB-U21 and refractory to enDUB-O4 and enDUB-Tr (Fig. 5H, I).

In cells co-expressing KCNQ1-YFP + ITCH, all five enDUBs again effectively decreased the amount of ubiquitin pulled-down with the channel (Fig. 6A–C). KCNQ1-YFP abundance in the presence of ITCH was increased by enDUB-Cz, enDUB-Tr, and enDUB-U21, while enDUB-O1 and enDUB-O4 had minimal impact (Fig. 6D, E). KCNQ1 surface density and whole-cell currents suppressed by ITCH was strongly increased by enDUB-O1, enDUB-Cz, and enDUB-U21, while enDUB-Tr

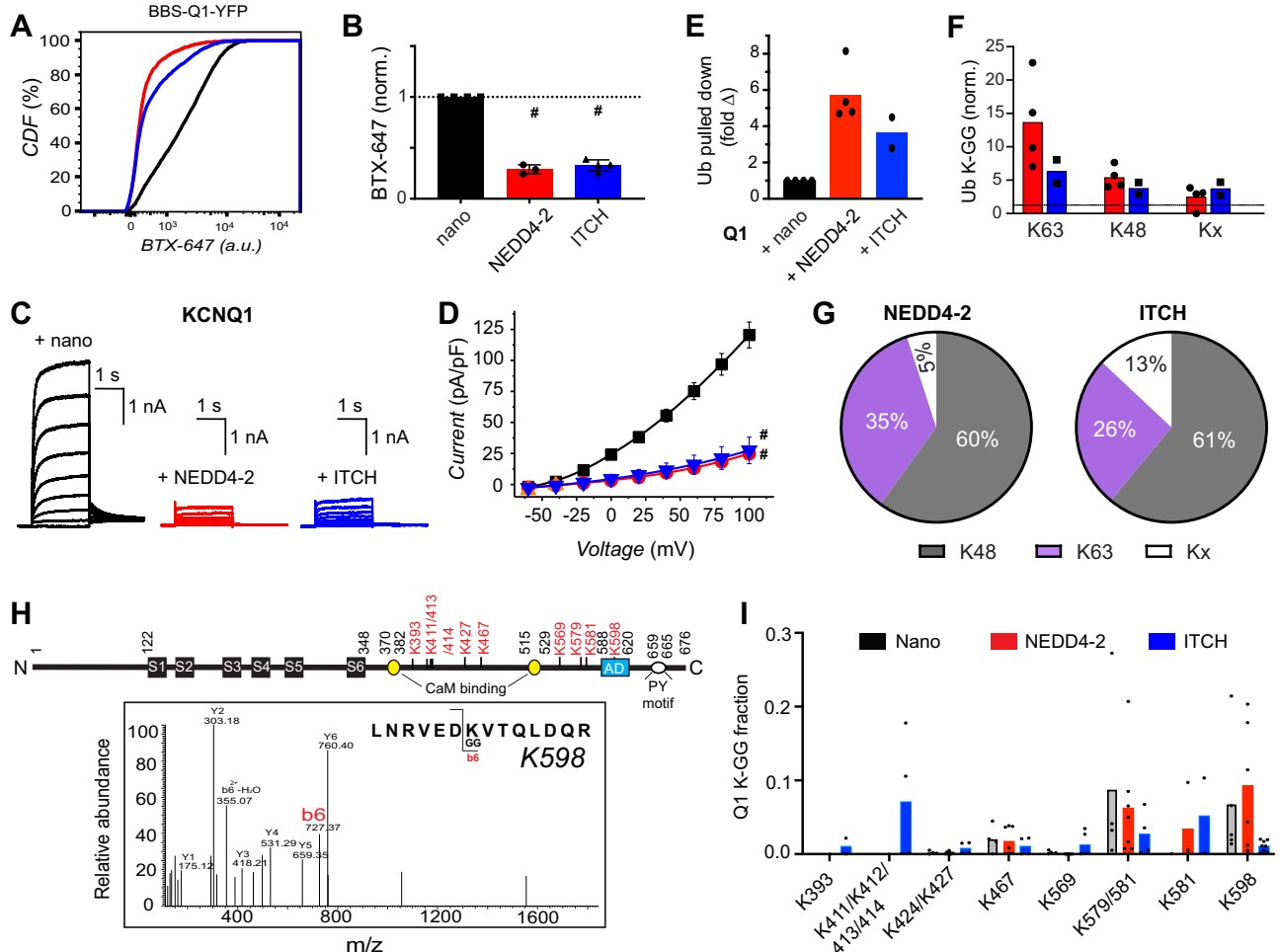

**Fig. 4 | KCNQ1 surface density is downregulated by the E3 ligases NEDD4-2 and ITCH with distinct polyubiquitin signatures. A** Representative flow cytometry CDF plots showing surface channel density (BTX-647 fluorescence) in cells expressing BBS-KCNQ1-YFP with nano (black), NEDD4-2 (red) or ITCH (blue). **B** Mean BBS-KCNQ1-YFP surface expression (BTX-647 fluorescence) analyzed from YFP-positive cells ($N = 3$-4; $\#p < 0.0001$, one-way ANOVA and Dunnett's multiple comparisons test: $\#p < 0.0001$, nano vs NEDD4-2 and $\#p < 0.0001$, nano vs ITCH). Data were normalized to values from the nano control group (dotted line) and presented as mean ± SD. **C** Exemplar KCNQ1 current traces from whole-cell patch clamp measurements in CHO cells. **D** Population *I-V* curves for KCNQ1 + nano (control; black, $n = 14$), KCNQ1 + NEDD4-2 (red, $n = 9$), and KCNQ1 + ITCH (blue, $n = 11$) ($\#p < 0.0001$, two-way ANOVA, with Tukey's multiple comparisons: $\#p < 0.0001$, nano vs NEDD4-2 and $\#p < 0.0001$, nano vs ITCH). Data are presented as mean values ± SEM. **E** Quantification of ubiquitin that was pulled down with

KCNQ1-YFP expressed with nano (control, $n = 4$), NEDD4-2 ($n = 4$) or ITCH ($n = 2$), as assessed by mass spectrometry. '*n*' is the number of monomeric and dimeric KCNQ1 bands analyzed across $N = 2$ independent experiments. Data were normalized to the control group and represented as the mean value. **F** Mass spectrometric evaluation of ubiquitin lysine residues that are di-gly modified (number of monomeric and dimeric KCNQ1 bands analyzed $n = 4$ across $N = 2$ independent experiments). Data were normalized to the control group. **G** Fractional distribution of Ub linkages as assessed by mass spectrometric analysis of immunoprecipitated and gel excised bands of KCNQ1-YFP co-expressed with NEDD4-2 or ITCH. **H** *Top*, cartoon of KCNQ1 with di-gly-modified lysine residues identified by mass spectrometry shown in red. *Bottom*, representative MS2 spectra trace of KCNQ1 peptide with di-gly modification of K598. **I** Quantification of di-gly-modified lysine residues on KCNQ1 peptides ($N = 2$ independent experiments). Source data are provided as a Source Data file.

yielded an intermediate rescue (Fig. 6G-I). The partial effect of enDUB-Tr in KCNQ1 channels downregulated by ITCH contrasted sharply with the lack of an effect observed with NEDD4-2.

Overall, the pattern of effects of distinct enDUBs on KCNQ1 surface density and currents differs among the basal, NEDD4-2 and ITCH overexpression conditions, emphasizing the adjustability of the ubiquitin code in response to changes in the cellular context, and further affirming the utility of linkage-selective enDUBs as tools to decipher the polyubiquitin code regulation of protein substrates in situ.

### Differential impact of linkage-selective enDUBs on KCNQ1 total and surface expression in cardiomyocytes

The malleability of the polyubiquitin code regulation of KCNQ1 revealed by our experiments suggested that polyubiquitin linkage regulation of the channel could potentially vary substantively in

different cell types. As such, it was of interest to probe the putative role of distinct polyubiquitin chain linkages on total and surface expression of KCNQ1 within the native context of ventricular cardiomyocytes. We generated adenoviral vectors expressing BBS-KCNQ1-YFP and the different enDUBs in a bicistronic IRES-mCherry format, and used these to infect acutely cultured adult guinea pig ventricular cardiomyocytes. We utilized confocal microscopy to visualize total channel abundance (YFP fluorescence), nano or enDUB expression (mCherry fluorescence), and surface channels (Alexa fluor 647 fluorescence) (Fig. 7A).

Control cells expressing BBS-Q1-YFP + nano-IRES-mCherry displayed robust YFP, mCherry, and 647 fluorescence signals indicating a fraction of the channels were present at the cardiomyocyte surface (Fig. 7B–D). Intriguingly, enDUB-O1 and enDUB-U21 strongly boosted KCNQ1 abundance (YFP fluorescence), enDUB-Cz had an intermediate effect, while enDUB-O4 and enDUB-Tr had no impact (Fig. 7B, C).

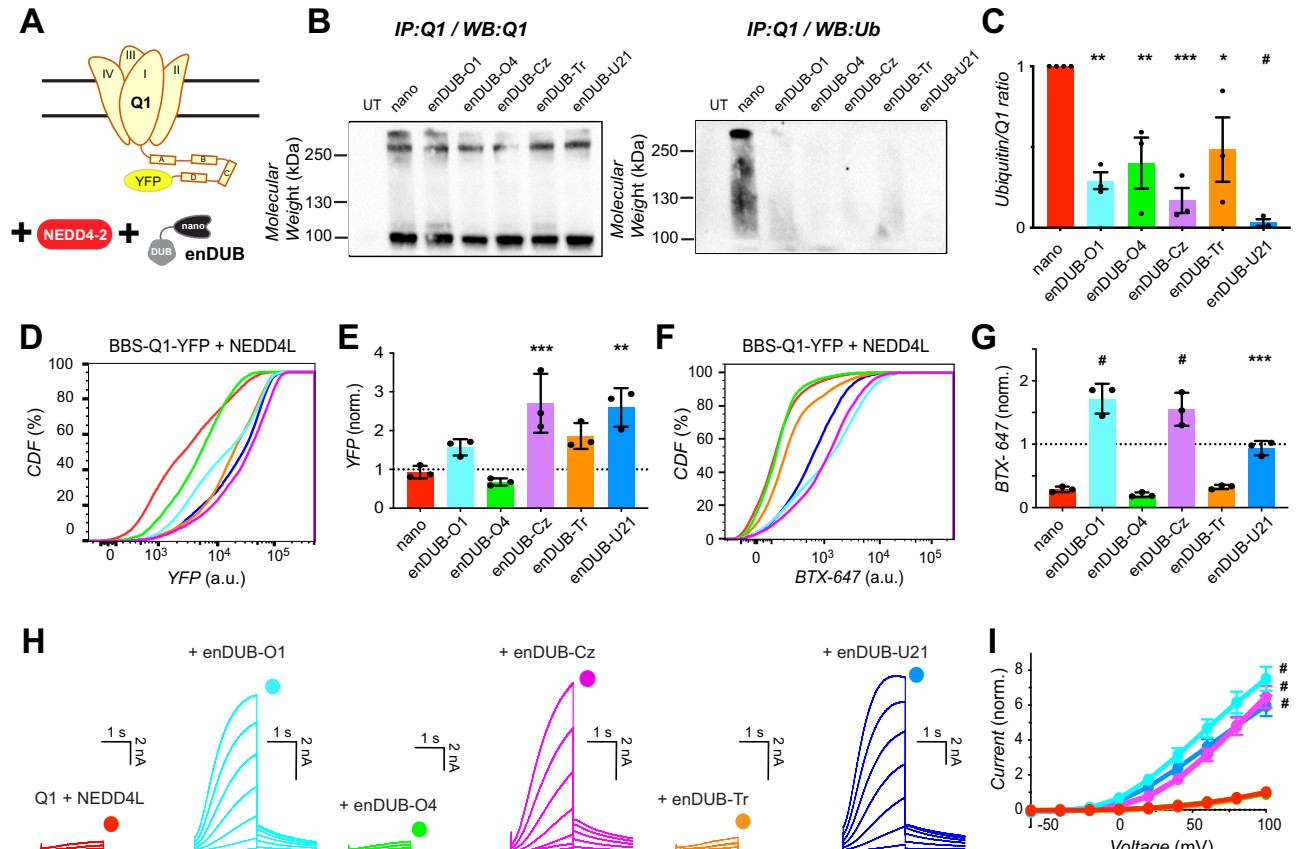

**Fig. 5 | Differential impact of distinct linkage-selective enDUBs on reversing KCNQ1 functional downregulation by NEDD4-2. A** Experimental design schematic; BBS-KCNQ1-YFP is co-expressed with NEDD4-2 and either nano (control) or an enDUB in HEK293 cells. **B** *Left*, Western blot of KCNQ1-YFP pulldowns from cells expressing KCNQ1-YFP + NEDD4-2 and either nano or the indicated enDUB, probed with anti-KCNQ1 antibody. *Right*, same blot stripped and probed with anti-ubiquitin. **C** Relative KCNQ1 ubiquitination computed by the ratio of anti-ubiquitin/anti-KCNQ1 signal intensities. Data are presented as mean values ± SEM; ($N$ = 3 biological replicates); ***$p$ = 0.0002, one-way ANOVA with Dunnett's multiple comparisons: **$p$ = 0.0010, nano vs enDUB-O1; **$p$ = 0.0041, nano vs enDUB-O4; ***$p$ = 0.0002, nano vs enDUB-Cz; *$p$ = 0.0123, nano vs enDUB-Tr; #$p$ < 0.0001, nano vs enDUB-U21. **D** Representative flow cytometry CDF plots showing total channel expression (YFP fluorescence) in cells expressing BBS-KCNQ1-YFP + NEDD4-2 and either nano or an enDUB. **E** Mean KCNQ1 expression analyzed from YFP- and CFP-positive cells. Data were normalized to values from the nano control group (dotted line) and presented as mean values ± SEM ($N$ = 3; ***$p$ = 0.0002, one-way ANOVA with Dunnett's multiple comparisons: $p$ = 0.2619, nano vs enDUB-O1; $p$ = 0.9039,

nano vs enDUB-O4; ***$p$ = 0.0008, nano vs enDUB-Cz; $p$ = 0.0624, nano vs enDUB-Tr; **$p$ = 0.0014, nano vs enDUB-U21). **F** Representative CDF plots showing surface channels (BTX-647 fluorescence) in cells expressing BBS-KCNQ1-YFP + NEDD4-2 and either nano or an enDUB. **G** Mean BBS-KCNQ1-YFP surface expression analyzed from YFP- and CFP- positive cells. Data were normalized to values from the nano control group (dotted line) and presented as mean values ± SEM ($N$ = 3; #$p$ < 0.0001, one-way ANOVA with Dunnett's multiple comparisons: #$p$ < 0.0001, nano vs enDUB-O1; $p$ = 0.9328, nano vs enDUB-O4; #$p$ < 0.0001, nano vs enDUB-Cz; $p$ = 0.9979, nano vs enDUB-Tr; ***$p$ = 0.0009, nano vs enDUB-U21). **H** Exemplar KCNQ1 + KCNE1 currents from whole-cell patch clamp measurements in CHO cells. **I** Population *I-V* curves for nano (control; red, $n$ = 18), enDUB-O1 (turquoise, $n$ = 9), enDUB-O4 (green, $n$ = 9), enDUB-Cz (pink, $n$ = 10), enDUB-Tr (orange, $n$ = 10) and enDUB-U21 (blue, $n$ = 8). Data are presented as mean values ± SEM; #$p$ < 0.0001, two-way ANOVA, with Tukey's multiple comparisons: #$p$ < 0.0001, nano vs enDUB-O1; $p$ > 0.9999, nano vs enDUB-O4; #$p$ < 0.0001, nano vs enDUB-Cz; $p$ > 0.9999, nano vs enDUB-Tr; #$p$ < 0.0001, nano vs enDUB-U21. Source data are provided as a Source Data file.

Channel surface density was strongly enhanced by enDUB-O1 while enDUB-Cz and enDUB-U21 had intermediate effects. By contrast, enDUB-O4 decreased surface density while enDUB-Tr had no impact (Fig. 7B, D).

Thus, the signature pattern of distinct enDUBs effects on KCNQ1 expression and surface density in cardiomyocytes differs qualitatively and quantitatively from observations in HEK293 cells, further demonstrating the cell-context-dependent plasticity of the ubiquitin code.

**KCNQ1 disease mutants alter signature pattern of responsiveness to distinct enDUBs**
Loss-of-function mutations in KCNQ1 that perturb channel trafficking cause LQT1 (Long QT Syndrome type 1) and may also lead to deafness. We previously showed that targeted deubiquitination with an enDUB can rescue channel surface density and ionic currents in a subset of

trafficking-deficient KCNQ1 mutations[29]. However, it is unknown whether such KCNQ1 missense mutations alter polyubiquitin linkage regulation of the channel. Accordingly, we tested the impact of the different enDUBs on abndance and surface density of four individual LQT1 mutations localized to the KCNQ1 C terminus− R366W, R539W, T587M and G589D (Fig. 8A). The mutations mostly had relatively modest impact on KCNQ1 abundance, with only T587M yielding a 30% decrease in YFP fluorescence intensity (Fig. 8F). The signature pattern of enDUB effects on expression levels of the mutants was similar to wild type KCNQ1 under basal conditions−enDUB-Cz, enDUB-Tr, and enDUB-U21 all increased YFP fluorescence whereas enDUB-O1 and enDUB-O4 had minimal effects (Fig. 8B, D, F, H).

Measuring surface density indicated the four mutations caused varying degrees of channel retention, ranging from moderate (R66W; 20% reduction) to intermediate (R539W; 60% reduction), and severe (T587M and G589D; 80% each) (Fig. 8C, E, G, I). The enDUBs had

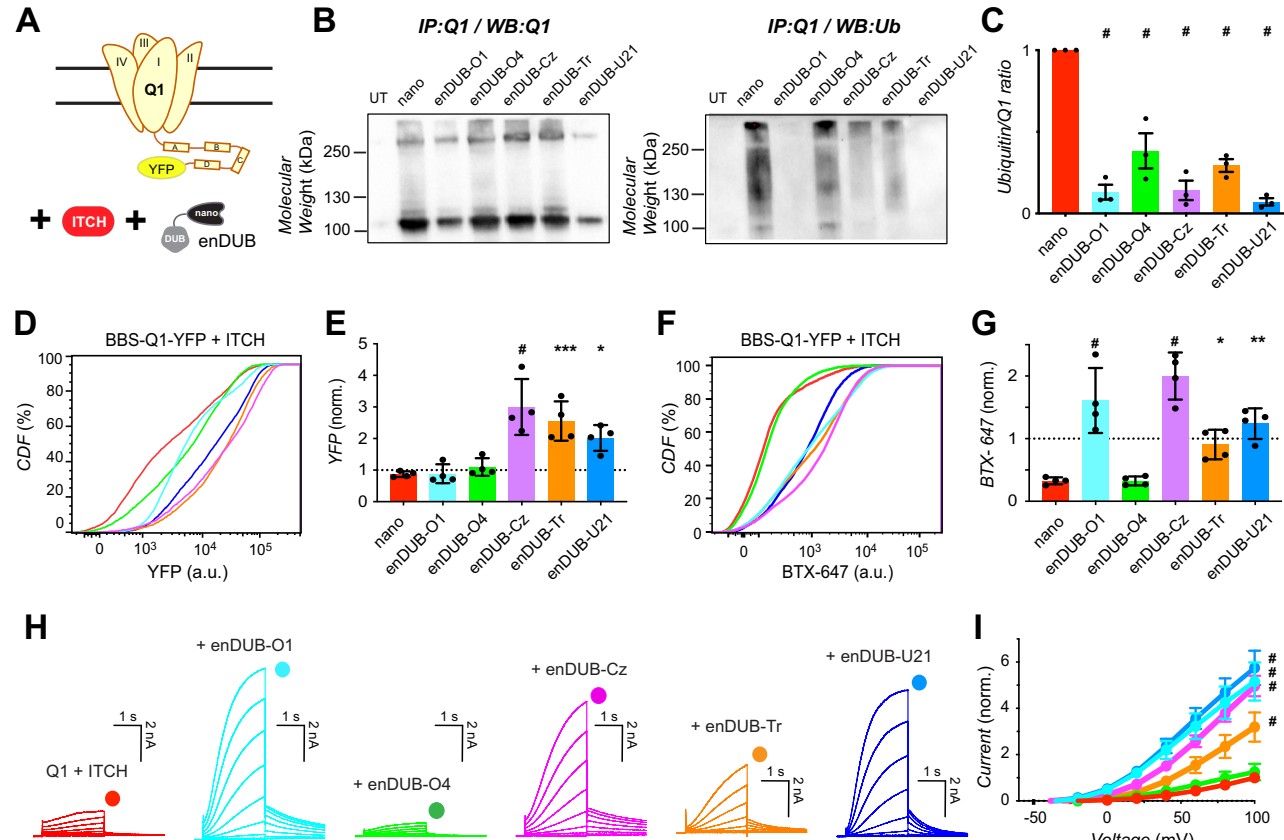

**Fig. 6 | Differential impact of distinct linkage-selective enDUBs on reversing KCNQ1 functional downregulation by ITCH. A** Experimental design schematic; BBS-KCNQ1-YFP is co-expressed with ITCH and either nano (control) or an enDUB in HEK293 cells. **B** *Left*, Western blot of KCNQ1-YFP pulldowns from cells expressing KCNQ1-YFP + ITCH and either nano or the indicated enDUB, probed with anti-KCNQ1 antibody. *Right*, same blot stripped and probed with anti-ubiquitin. **C** Relative KCNQ1 ubiquitination computed by the ratio of anti-ubiquitin/anti-KCNQ1 signal intensities. Data are presented as mean values ± SEM; $N = 3$ biological replicates; #$p < 0.0001$, one-way ANOVA with Dunnett's multiple comparisons: #$p < 0.0001$, nano vs every enDUB. **D** Representative flow cytometry CDF plots showing total channel expression (YFP fluorescence) in cells expressing BBS-KCNQ1-YFP + ITCH and either nano or an enDUB. **E** Mean KCNQ1 expression analyzed from YFP- and CFP- positive cells ($N = 3$; #$p < 0.0001$, one-way ANOVA with Dunnett's multiple comparisons: $p > 0.9999$, nano vs enDUB-O1; $p = 0.9479$, nano vs enDUB-O4; #$p < 0.0001$, nano vs enDUB-Cz; ***$p = 0.0007$, nano vs enDUB-Tr; *$p = 0.0189$, nano vs enDUB-U21). Data were normalized to values from the nano

control group (dotted line) and presented as mean values ± SEM. **F** Representative CDF plots showing surface channels (BTX-647 fluorescence) in cells expressing BBS-KCNQ1-YFP + NEDD4-2 and either nano or an enDUB. **G** Mean BBS-KCNQ1-YFP surface expression analyzed from YFP- and CFP- positive cells ($N = 3$; #$p < 0.0001$, one-way ANOVA with Dunnett's multiple comparisons: #$p < 0.0001$, nano vs enDUB-O1; $p = 0.9928$, nano vs enDUB-O4; #$p < 0.0001$, nano vs enDUB-Cz; *$p = 0.0265$, nano vs enDUB-Tr; **$p = 0.0012$, nano vs enDUB-U21). Data were normalized to values from the nano control group (dotted line) and presented as mean values ± SEM. **H** Exemplar KCNQ1 + KCNE1 currents from whole-cell patch clamp measurements in CHO cells. **I** Population *I-V* curves for nano (control; red, $n = 18$), enDUB-O1 (turquoise, $n = 9$), enDUB-O4 (green, $n = 8$), enDUB-Cez (pink, $n = 10$), enDUB-Tr (orange, $n = 10$) and enDUB-U21 (blue, $n = 8$). Data are presented as mean values ± SEM; #$p < 0.0001$, two-way ANOVA with Tukey's multiple comparisons: #$p < 0.0001$, nano vs enDUB-O1; $p = 0.7895$, nano vs enDUB-O4; #$p < 0.0001$, nano vs enDUB-Cz; #$p < 0.0001$, nano vs enDUB-Tr; #$p < 0.0001$, nano vs enDUB-U21. Source data are provided as a Source Data file.

distinctive signature patterns on rescue of surface density: R366W surface density was increased by enDUB-Tr and decreased by enDUB-O4 (Fig. 8C); R539W was moderately rescued by enDUB-Tr and enDUB-Cz; T587M and G589D were most strongly rescued by enDUB-O1 with smaller effects produced by enDUB-Cz, enDUB-Tr, and enDUB-U21. Thus, these missense mutations in KCNQ1 result in unique changes in polyubiquitin linkage code regulation of channel subcellular localization that potentially contributes to the molecular basis of disease.

## Discussion

Ubiquitination is an intricate posttranslational modification that regulates the abundance, localization, and function of all proteins to control biology. The ubiquitin code refers to the complex system of ubiquitin writers, erasers, editors, and readers that govern the nature of ubiquitin signals on individual proteins and how they are decoded in cells to yield diverse functional outcomes[1,5,11]. Understanding the ubiquitin code regulation of proteins is an active research frontier with broad implications for insights into physiology, disease mechanisms, and drug

development. The prevalence of polyubiquitin chains with distinct linkages and topologies on proteins is a prominent aspect of the ubiquitin code[1,3,5,6]. However, the functions of diverse polyubiquitin signatures on most proteins is unclear due to a lack of approaches to specifically tune polyubiquitin linkages on individual proteins in live cells. In this work we describe a method to address this core deficiency by developing linkage-selective enDUBs and apply them to investigate polyubiquitin regulation of KCNQ1-YFP abundance, subcellular localization, and function in live cells. The results validate the utility of linkage-selective enDUBs to help decipher the polyubiquitin code regulation of protein function and expression in different cellular contexts, and provide insights into divergent regulation of surface expression of an ion channel by distinct polyubiquitin linkages.

### Validity of enDUBs as a tool to decode polyubiquitin linkage function in cells

The strength of the conclusions drawn from the observed data hinges critically on the capacity of putative linkage-selective enDUBs to

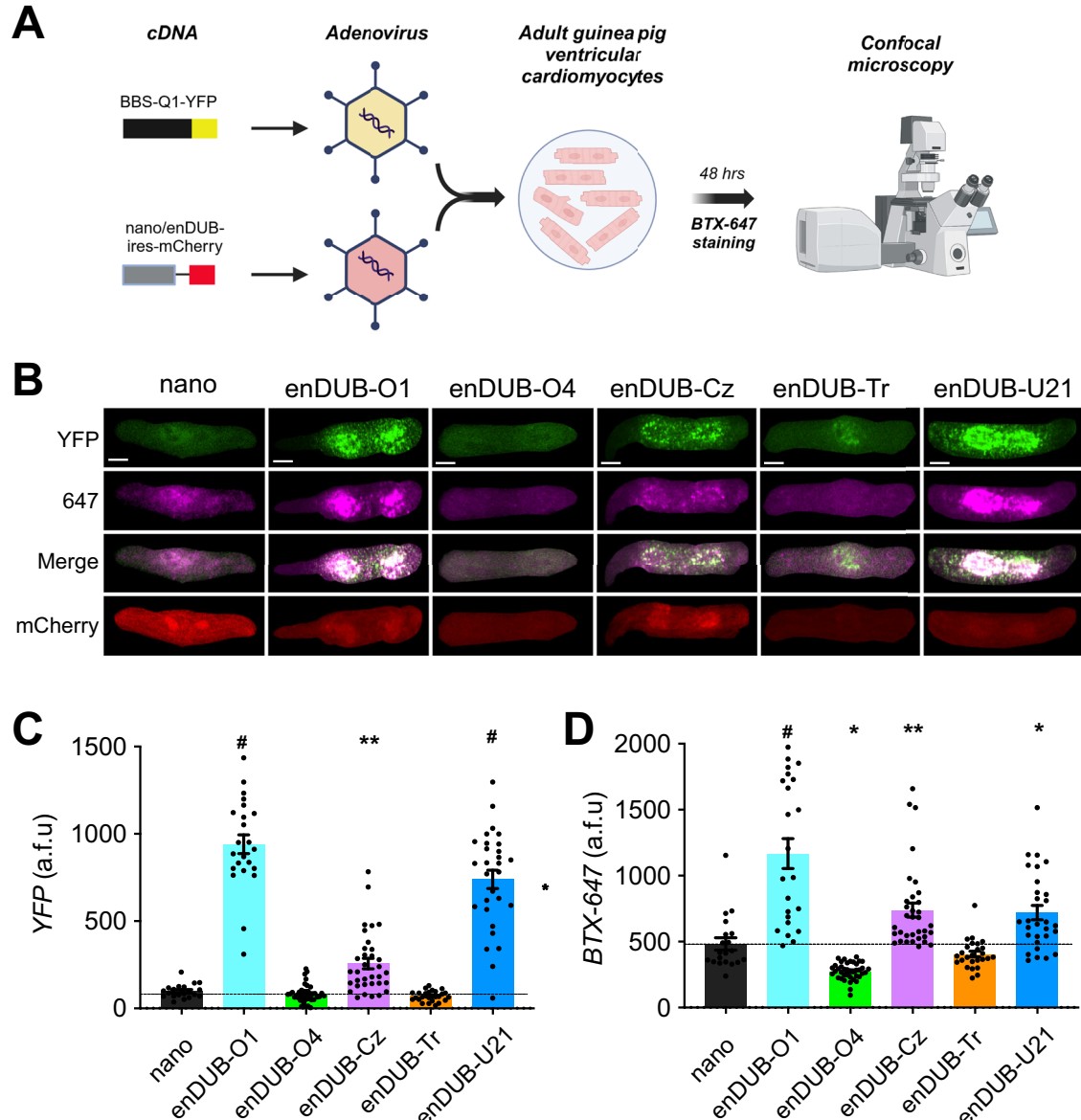

**Fig. 7 | Differential impact of linkage-selective enDUBs on KCNQ1 abundance and surface density in cardiomyocytes. A** Schematic of the experimental design. Created in BioRender. Colecraft, H. (https://BioRender.com/8ncfwc8)
**B** Representative confocal images (maximum intensity Z-projection) of adult guinea pig ventricular cardiomyocytes expressing BBS-KCNQ1-YFP with either nano-IRES-mCherry or an enDUB-IRES-mCherry, showing YFP (top row), BTX-647 (second row), YFP and BTX-647 overlay (third row), and mCherry (bottom) fluorescence signals. Scale bar, 10 μm. **C** Quantification of YFP fluorescence intensity (total channel expression) measured from confocal images (N = 3 isolations; mean and

SEM; #p < 0.0001, one-way ANOVA and Dunnett's multiple comparisons test: #p < 0.0001, nano vs enDUB-O1; p = 0.9964, nano vs enDUB-O4; **p = 0.0054, nano vs enDUB-Cz; p = 0.9666, nano vs enDUB-Tr; #p < 0.0001, nano vs enDUB-U21).
**D** Quantification of the surface BTX-647 fluorescence intensity (surface channels) measured from confocal images (N = 3 isolations; mean and SEM; #p < 0.0001, one-way ANOVA and Dunnett's multiple comparisons test: #p < 0.0001, nano vs enDUB-O1; *p = 0.0346, nano vs enDUB-O4; **p = 0.0072, nano vs enDUB-Cz; p = 0.7954, nano vs enDUB-Tr; *p = 0.0189, nano vs enDUB-U21). Source data are provided as a Source Data file.

specifically hydrolyze intended polyubiquitin chains in a substrate selective manner. The approach was inspired by previous elegant work demonstrating the capacity of certain ovarian tumor (OTU) DUB catalytic domains to hydrolyze polyubiquitin chains in a linkage-selective manner in vitro[26,27]. Nevertheless, a potential concern was that the linkage-selectivity of polyubiquitin chain hydrolysis would not hold in situ, or be compromised by prolonged tethering of DUB to substrate enabled by a nanobody. All four putative linkage-selective enDUBs (enDUB-O1, enDUB-O4, enDUB-Cz and enDUB-Tr) as well as the non-selective enDUB-U21 effectively reduced the amount of ubiquitin using a pan-ubiquitin antibody on KCNQ1-YFP. While reassuring in terms of confirming the in vivo enzymatic viability of the distinct enDUBs, this

biochemical result in itself could not address whether the enDUBs were behaving in a linkage-selective manner. Rather, their distinctive impact on KCNQ1-YFP abundance, subcellular localization, and functional currents provides the strongest evidence that their enzymatic uniqueness is maintained in situ. A previous study examined the impact of K63 polyubiquitin linkages on post-endocytic sorting of epidermal growth factor receptor (EGFR) by fusing the K63 linkage-selective DUB, AMSH (associated molecule with the Src homology 3 domain of signal transducing adapter molecule), to the receptor. Mass spectrometry experiments suggested linkage-preference of AMSH for K63 over K48 hydrolysis was maintained even in this most extreme configuration where the DUB is covalently attached to the substrate[31].

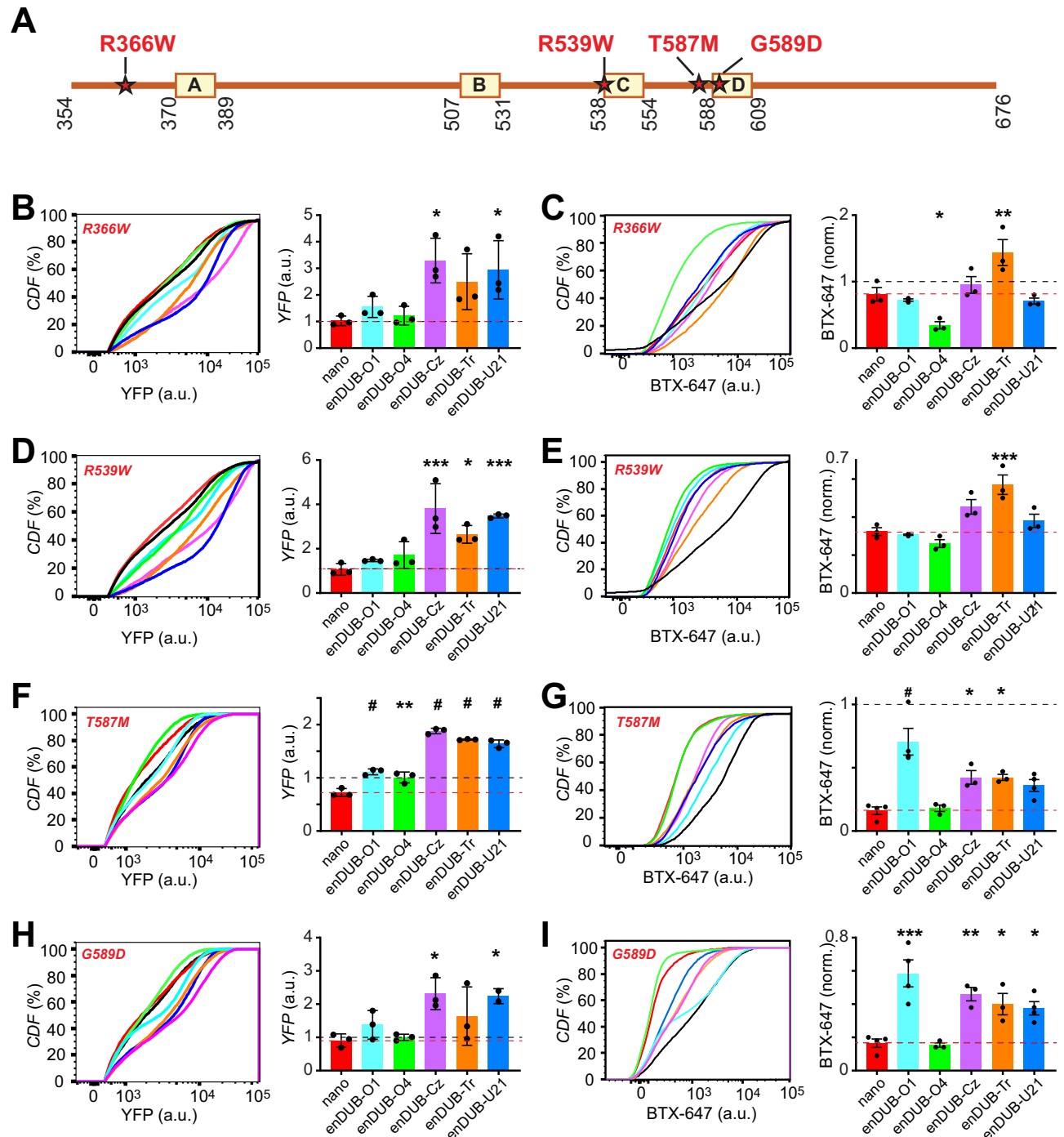

Our findings with the enDUBs are consistent with this previous result in terms of indicating that linkage-selective DUBs preserve their ubiquitin linkage preferences when tethered to a substrate. Nevertheless, enDUBs provide a clear advantage in versatility and generalizability as they can be adapted to use for any endogenous protein substrate without the need for covalent attachment of the DUB catalytic domain.

How do linkage-selective enDUBs contrast with prevailing methods to examine functions of distinct polyubiquitin linkages in vivo? A popular approach is to express K-to-R ubiquitin mutants in cells or organisms and observe the functional output. For example, this approach has been used effectively to examine global consequences of eliminating particular ubiquitin linkages in yeast[32], and to validate hypothesized functional roles on individual proteins in cells[33–35]. However, this approach alters ubiquitination of the entire cellular

proteome in contrast to the substrate-specific nature of the linkage-selective enDUB method. Overall, the linkage-selective enDUB strategy uniquely fills a gap in the functional analyses of polyubiquitin linkages in situ, and is complementary to prevailing methods for addressing this aspect of the ubiquitin code[36].

## Insights into polyubiquitin linkage regulation of KCNQ1 abundance

In HEK293 cells, KCNQ1-YFP expression was upregulated by enDUB-Cz, enDUB-Tr, and enDUB-U21, but not enDUB-O1 or enDUB-O4. These results indicate a dominant contribution of K11 and K29/K33 but not K63 or K48 to KCNQ1-YFP degradation in this cellular context (Fig. 9). This result was unexpected because K48 was the most abundant linkage type associated with KCNQ1 and is the canonical polyubiquitin

**Fig. 8 | KCNQ1 disease mutants alter signature pattern of responsiveness to distinct enDUBs. A** Scheme of LQT1 mutations mapped on a cartoon of the KCNQ1 C-terminus. **B** *Left*, representative flow cytometry CDF plots of YFP fluorescence (channel total expression) in cells expressing BBS-KCNQ1[R366W]-YFP and either nano or an enDUBs. *Right*, mean values ± SEM of channel expression analyzed from YFP- and CFP- positive cells ($N = 3$ independent experiments). Data were normalized to the values from WT BBS-KCNQ1-YFP + nano control group (dotted line, black). Red dotted line corresponds to the values of mutant BBS-KCNQ1-YFP + nano. *$p = 0.0114$, one-way ANOVA and Dunnett's multiple comparisons test: $p = 0.8543$, nano vs enDUB-O1; $p = 0.9973$, nano vs enDUB-O4; *$p = 0.0117$, nano vs enDUB-Cz; $p = 0.1149$, nano vs enDUB-Tr; *$p = 0.0321$, nano vs enDUB-U21. **C** *Left*, representative CDF plots showing BTX-647 fluorescence (surface channels) in cells expressing BBS-KCNQ1[R366W]-YFP and either nano or an enDUB. *Right*, mean values ± SEM of channel expression analyzed from YFP- and CFP- positive cells ($N = 3$). Data were normalized to values from WT BBS-Q1-YFP+nano control group (black dotted line). ***$p = 0.0003$, one-way ANOVA and Dunnett's multiple comparisons test: $p = 0.9665$, nano vs enDUB-O1; *$p = 0.0372$, nano vs enDUB-O4; $p = 0.8216$, nano vs enDUB-Cz; **$p = 0.0058$, nano vs enDUB-Tr; $p = 0.9345$, nano vs enDUB-U21. **D** Data for BBS-KCNQ1[R539W]-YFP; same format as B. ***$p = 0.0002$, one-way ANOVA and Dunnett's multiple comparisons test: $p = 0.8416$, nano vs enDUB-O1; $p = 0.5072$, nano vs enDUB-O4; ***$p = 0.0003$, nano vs enDUB-Cz;

*$p = 0.0184$, nano vs enDUB-Tr; ***$p = 0.0008$, nano vs enDUB-U21. **E** Data for BBS-KCNQ1[R539W]-YFP; same format as C. ***$p = 0.0002$, one-way ANOVA and Dunnett's multiple comparisons test: $p = 0.9949$, nano vs enDUB-O1; $p = 0.5228$, nano vs enDUB-O4; $p = 0.0538$, nano vs enDUB-Cz; ***$p = 0.0007$, nano vs enDUB-Tr; $p = 0.6163$, nano vs enDUB-U21. **F** Data for BBS-KCNQ1[T587M]-YFP; same format as B. #$p < 0.0001$, one-way ANOVA and Dunnett's multiple comparisons test: #$p < 0.0001$, nano vs enDUB-O1; **$p = 0.001$, nano vs enDUB-O4; #$p < 0.0001$, nano vs enDUB-Cz; #$p < 0.0001$, nano vs enDUB-Tr; #$p < 0.0001$, nano vs enDUB-U21. **G** Data for BBS-KCNQ1[T587M]-YFP; same format as C. ***$p = 0.0001$, one-way ANOVA and Dunnett's multiple comparisons test: #$p < 0.0001$, nano vs enDUB-O1; $p = 0.9995$, nano vs enDUB-O4; *$p = 0.0344$, nano vs enDUB-Cz; *$p = 0.0344$, nano vs enDUB-Tr; $p = 0.096$, nano vs enDUB-U21. **H** Data for BBS-KCNQ1[G589D]-YFP; same format as B. *$p = 0.02$, one-way ANOVA and Dunnett's multiple comparisons test: $p = 0.6523$, nano vs enDUB-O1; $p = 0.9993$, nano vs enDUB-O4; *$p = 0.0166$, nano vs enDUB-Tr; *$p = 0.042$, nano vs enDUB-U21. **I** Data for BBS-KCNQ1[G589D]-YFP; same format as C. ***$p = 0.0003$, one-way ANOVA and Dunnett's multiple comparisons test: ***$p = 0.0003$, nano vs enDUB-O1; $p = 0.9999$, nano vs enDUB-O4; **$p = 0.0051$, nano vs enDUB-Cz; *$p = 0.0228$, nano vs enDUB-Tr; *$p = 0.0428$, nano vs enDUB-U21. Source data are provided as a Source Data file.

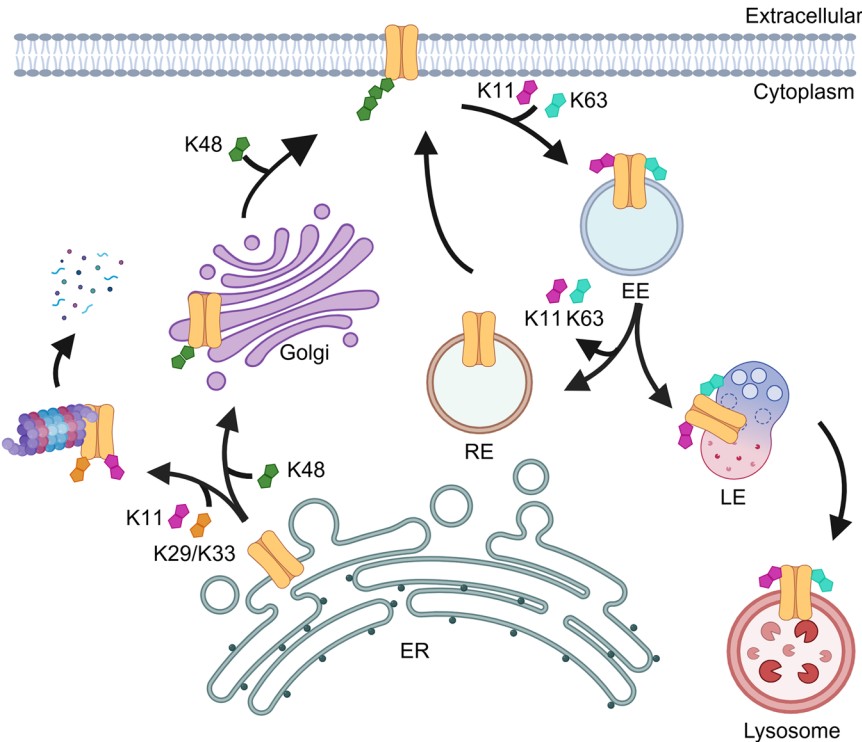

**Fig. 9 | Differential roles of distinct polyubiquitin linkages in regulating KCNQ1 expression and trafficking in HEK293 cells.** Cartoon showing proposed differential roles of distinct polyubiquitin chains in regulating KCNQ1 expression and trafficking among subcellular compartments in HEK 293 cells under basal conditions. This polyubiquitin code for KCNQ1 is not immutable and can be changed by both extrinsic (changing cellular conditions or cell types) and intrinsic (mutations in the substrate) factors. Created in BioRender. Shanmugam, S. (https://BioRender.com/w3nu0kx).

signal associated with protein degradation[1,5,37,38]. Precedence for K48 polyubiquitination not resulting in protein degradation is provided by the yeast transcription factor, Met4, which was found to encode a ubiquitin interaction motif (UIM) that intramolecularly binds K48 polyubiquitin and protects the protein from proteasomal degradation[39]. Our results give reason to wonder whether KCNQ1 may similarly encode an intrinsic UIM that protects the channel from K48 polyubiquitin-mediated proteasomal degradation. Future use of linkage-selective enDUBs may uncover that this phenomenon of non-degradative roles for K48 chains is more prevalent than currently appreciated.

Another surprise was the apparently dominant role of minor polyubiquitin chains associated with KCNQ1-YFP, K11 and K29/K33, in mediating channel degradation. Previous studies have linked both these chain types to protein degradation pathways; K11 chains assembled by the anaphase-promoting complex (APC/C) mediate degradation of cell cycle regulators[35,40], and K29 chains have also been associated with protein degradation[41]. Further, both K11- and K29-ubiquitin chains have been associated with endoplasmic reticulum-associated degradation (ERAD)[42,43]. The use of an apparently minor polyubiquitin species to control protein elimination may act as an effective valve to control the rate of channel degradation. A caveat is

that our measure of the fraction of atypical polyubiquitin linkage types associated with the channel may be an underestimate because the mass spectrometry analyses was constrained to visually identified monomers and dimers of the channel from Coomassie-stained gels. Given that extensive polyubiquitination can result in substantial increases in molecular weight, the contribution of potential highly polyubiquitinated KCNQ1-YFP species would be underrepresented in the mass spectrometry data.

In contrast to HEK293 cells, KCNQ1-YFP abundance in adult guinea pig ventricular cardiomyocytes was robustly upregulated by enDUB-O1 and enDUB-U21, moderately by enDUB-Cz, downregulated by enDUB-O4 and unaffected by enDUB-Tr. These variations are consistent with fundamental differences in the ubiquitin code regulation of KCNQ1-YFP expression between HEK293 cells and cardiomyocytes. While K63 polyubiquitin is conventionally ascribed non-degradative functions, our results suggest a prominent role of K63 polyubiquitin in mediating degradation of KCNQ1-YFP in cardiomyocytes. One possibility is that lysosomal degradation of the channel plays a larger role in cardiomyocytes than we observed in HEK293 cells. A previous report indicated K63 seeds formation of K48/K63 branched ubiquitin chains that enable proteasomal degradation of certain substrates[44]. While the lack of effect of enDUB-O4 on KCNQ1-YFP abundance in cardiomyocytes does not support a role for K48 polyubiquitin in this context, it is possible that K63 may seed other types of branched ubiquitin chains that underlie degradation of the channel in heart cells.

### Insights into polyubiquitin linkage regulation of KCNQ1 subcellular localization and surface trafficking

The plasma membrane is the site of KCNQ1 function, making the impact of enDUBs on channel surface density particularly important. In HEK293 cells, the relative effects of the distinct enDUBs on KCNQ1 surface density varied substantially under different conditions, contrasting with the more stable impact on channel abundance. EnDUB-O1 and enDUB-U21 had no effect on channel surface density under baseline conditions but effectively rescued KCNQ1 surface density that had been severely depressed by NEDD4-2 or ITCH; enDUB-Cz increased surface density under all conditions; and enDUB-O4 itself depressed KCNQ1 surface density under baseline conditions. Combined with the distinctive impact of these enDUBs on KCNQ1 subcellular localization and rates of forward trafficking and endocytosis, the results are consistent with K63, K11, and K29/K33 polyubiquitin chains mediating net intracellular retention of the channel, but achieved in different ways. K11 promotes ER retention and degradation, enhances endocytosis, and reduces channel recycling; K29/K33 primarily promotes ER retention and degradation; K63 enhances endocytosis and reduces channel recycling (Fig. 9). Previous studies have reported roles of K63 polyubiquitin in mediating sorting of membrane proteins to post-endocytic vesicular compartments[34]. These studies provide evidence of roles for K11 and putatively K29/K33 polyubiquitin in controlling the subcellular localization and trafficking of an ion channel.

The pattern of enDUB effects on KCNQ1-YFP surface density under baseline conditions in cardiomyocytes differed from HEK293 cells. In cardiomyocytes, KCNQ1-YFP surface density was strongly enhanced by enDUB-O1; moderately increased by enDUB-Cz and enDUB-U21; unaffected by enDUB-Tr; and decreased by enDUB-O4. This signature pattern is closest to HEK293 cells over-expressing NEDD4-2 and may indicate that the ubiquitin code in cardiomyocytes under basal conditions is oriented towards having more K63 polyubiquitin on KCNQ1 compared to HEK293 cells.

Intriguingly, presumed removal of K48 by enDUB-O4 decreased KCNQ1-YFP surface density in both HEK293 cells and cardiomyocytes under baseline conditions. In HEK293 cells, this was accompanied by an increased retention in the ER and Golgi compartments. The most parsimonious interpretation of this result is that K48 polyubiquitin plays an active positive role in forward trafficking of KCNQ1 in both HEK293 cells and cardiomyocytes.

### Implications for disease and therapeutics development

Mutations in ion channels that impair their cell surface density underlie devastating diseases including cystic fibrosis, lethal cardiac arrhythmias, epilepsy, and intellectual disability. Trafficking of wild type ion channels can also be compromised in disease conditions such as diabetes in ways that contribute to pathology[15,16]. We previously showed that targeted deubiquitination with enDUB-O1 could rescue trafficking deficient mutations in KCNQ1 and cystic fibrosis transmembrane regulator (CFTR) that cause LQT1 and CF, respectively[29]. Consistent with the idea that net intracellular retention of ion channels due to mutations can occur in various ways we found that distinct KCNQ1 LQT1 mutations were differentially rescued by linkage-selective enDUBs. Surface density of R366W and R539W were most robustly increased by enDUB-Tr while T587M and G589D were most responsive to enDUB-O1. These results agree with previous findings that mutations in ion channels can alter their ubiquitination status in a manner that interferes with their surface expression[45–48]. Our findings suggest that linkage-selective enDUBs could have utility in elucidating the precise trafficking deficits caused by distinct ion channel disease mutations, and in identifying the most efficacious polyubiquitin linkage type/s to target for rescue. Beyond enDUBs, divalent small molecule deubiquitinase targeting chimeras (DUBTACs) that recruit deubiquitinases to stabilize target proteins have also been developed[49]. Deepened understanding of the versatility of the ubiquitin code combined with the uniqueness of distinct DUBs will expand the range of diseases and facilitate rational development of therapeutics and tool compounds enabled by this induced proximity modality.

In summary, we have developed and applied linkage-selective enDUBs to dissect the role of distinct poly-ubiquitin chains in regulating KCNQ1 stability, subcellular localization, and function. A limitation of our study is we have tested the approach on a tagged channel that we heterologously expressed in either HEK293 cells or cardiomyocytes. Fortunately, the approach is modular and can be adapted to study the functions of diverse ubiquitin linkages on endogenous proteins with only the need to identify a suitable nanobody binder (or other antibody-mimetic) for the target. Nanobody binders can be readily generated by alpaca immunization and phage display technologies or by screening synthetic nanobody libraries[50,51]. Alternatively, computational protein design algorithms may be used to design binders based on the sequence of the target protein[52]. Future studies will expand the suite of linkage-selective enDUBs to include other linkage types and probe ubiquitin code regulation of endogenous proteins. Altogether, linkage-selective enDUBs provide a unique tool that is complementary to prevailing methods to elucidate the intricate and dynamic polyubiquitin code in vivo.

## Methods

### Ethical statement

This study uses adult ventricular cardiomyocytes isolated from guinea pig performed in accordance with the guidelines and approval of the Institutional Animal Care and Use Committee at Columbia University.

### Plasmids and adenoviral generation

The construction of enDUB-O1 and enDUB-21 has been previously described[29]. To generate the other enDUBs we used polymerase chain reaction (PCR) to amplify the coding sequence for a GFP nanobody (vhhGFP4; nano) and cloned it into a xx-P2A-CFP cassette-containing mammalian expression vector[24] using NheI and AflII sites. The catalytic domain of OTUD4 (residues 22-196) was amplified by PCR and cloned upstream of the GFP nanobody sequence to generate enDUB-O4-P2A-CFP. The sequences for the catalytic domains of Cezanne (residues 53-

446) and TRABID (residues 245–696) were amplified by PCR and cloned downstream of GFP nanobody sequence to generate enDUB-Cz-P2A-CFP and enDUB-Tr-P2A-CFP, respectively. Catalytically inactive enDUBs were generated by site-directed mutagenesis using the Quik-Change Lightning Site-Directed Mutagenesis kit (Stratagene) to introduce a point mutation at their catalytic cysteine residue.

KCNQ1 constructs were made as previously described[25]. Briefly, overlap extension PCR was used to fuse the sequence for enhanced yellow fluorescent protein (EYFP) in frame KCNQ1 C terminus. A 13-residue bungarotoxin binding site peptide (BBS) sequence was introduced between sequences for residues 148 and 149 in KCNQ1 extracellular S1–S2 loop using QuikChange Lightning Site-Directed Mutagenesis kit (Stratagene) according to the manufacturer's instructions. LQT1 mutations were introduced in the N and C termini of KCNQ1 via site-directed mutagenesis as previously described[25,29]. Plasmid vectors encoding NEDD4-2 (Addgene #27000)[53] and ITCH (Addgene #11427)[54] were gifts from Joan Massague and Allan Weismann, respectively.

Adenoviruses encoding BBS-KCNQ1-YFP, enDUB-O1-IRES-mCherry, enDUB-O4-IRES-mCherry, enDUB-Cz-IRES-mCherry, enDUB-Tr-IRES-mCherry, and enDUB-U21-IRES-mCherry were generated by Vector Builder.

## Cell culture and transfection

HEK293 and CHO cells were gifts from the laboratory of R. Kass (Columbia University). Cells were mycoplasma free, as determined by the MycoFluor Mycoplasma Detection kit (Invitrogen). Low-passage HEK293 cells were cultured at 37 °C in DMEM supplemented with 8% FBS and 100 mg/mL penicillin–streptomycin. For Western blot and flow cytometry experiments, transfection of HEK293 cells was accomplished using the calcium phosphate precipitation method. Briefly, plasmid DNA was mixed with 62 μl of 2.5 M CaCl2 and sterile deionized water (to a final volume of 500 μL). The mixture was added dropwise, with constant tapping, to 500 μL 2 × HEPES-buffered saline containing (in mM) 50 HEPES, 280 NaCl and 1.5 Na2HPO4 (pH 7.09). The resulting DNA–calcium phosphate mixture was incubated for 20 min at room temperature (RT) and then added dropwise to HEK293 cells (60–80% confluent). Cells were washed with $Ca^{2+}$-free PBS after 4–6 h and maintained in supplemented DMEM.

CHO cells were transiently transfected with desired constructs in 35-mm tissue culture dishes [KCNQ1 (0.5 μg) and nanobody–P2A– CFP (0.5 μg) or enDUB (0.5 μg)], in the presence or absence of E3 ligases (NEDD4-2 or ITCH; 0.5 μg) using X-tremeGENE HP (1:2 DNA:reagent ratio) according to the manufacturer's instructions (Roche).

## Cardiomyocyte isolation and transduction

Isolation of adult guinea pig cardiomyocytes was performed under the guidelines of the Columbia University Animal Care and Use Committee. Before isolation, glass bottom culture dishes were pre-coated with 10 μg/mL laminin (Gibco). Adult Hartley male guinea pigs of age 3-4 weeks old (Charles River) were deeply anesthetized with 5% isoflurane and their hearts excised and mounted on a Langendorff perfusion apparatus. Ventricular myocytes were isolated by perfusing the heart first with KH solution (in mM, 118 NaCl, 4.8 KCl, 1 CaCl2, 25 HEPES, 1.25 K2HPO4, 1.25 MgSO4, 11 glucose, 0.02 EGTA, pH 7.4), followed by perfusion with enzymatic digestion solution containing 0.3 mg/mL Collagenase Type 4 (Worthington), 12.5 μg/mL protease, and 0.05% BSA in $Ca^{2+}$-free KH buffer for 20 min. After digestion, 40 mL of a high-$K^+$ solution containing 120 mM potassium glutamate, 25 mM KCl, 10 mM HEPES, 1 mM MgCl2 and 0.02 mM EGTA, pH 7.4, was perfused through the heart. Cells were subsequently dispersed in the high-$K^+$ solution. Healthy rod-shaped myocytes were cultured in Medium 199 (Life Technologies) supplemented with 10 mM HEPES (Gibco), 1 × MEM non-essential amino acids (Gibco), 2 mM L-glutamine (Gibco), 20 mM D-glucose

(Sigma-Aldrich), 1% (vol/vol) penicillin–streptomycin–glutamine (Fisher Scientific), 0.02 mg/mL vitamin B12 (Sigma-Aldrich) and 5% (vol/vol) FBS (Life Technologies) to promote attachment to dishes. After 3 h, the culture medium was changed to Medium 199 with 1% serum but otherwise supplemented as described above. Cultures were maintained in humidified incubators at 37 °C with 5% CO2. Adenoviral vectors were added to the medium and incubated overnight, followed by washing with fresh medium the next day. Experiments with cardiomyocytes were performed 1–2 d later.

## Flow cytometry assays

Cell surface and total KCNQ1 channel expression were assayed by flow cytometry in live, transfected HEK293 cells as previously described[25,29,55]. Briefly, 48 h post-transfection, cells cultured in 12-well plates were gently washed with ice cold PBS containing $Ca^{2+}$ and $Mg^{2+}$ (in mM: 0.9 CaCl2, 0.49 MgCl2, pH 7.4), and then incubated for 30 min in blocking medium (DMEM with 3% BSA) at 4 °C. The cells were then incubated with 1 μM Alexa Fluor 647 conjugated α-bungarotoxin (BTX-647; Life Technologies) in DMEM/3% BSA on a rocker at 4 °C for 1 h, followed by washing three times with PBS (containing $Ca^{2+}$ and $Mg^{2+}$). Cells were gently harvested in $Ca^{2+}$-free PBS and assayed by flow cytometry using a BD LSRII Cell Analyzer (BD Biosciences, San Jose, CA, United States). CFP- and YFP-tagged proteins were excited at 407 and 488 nm, respectively, and Alexa Fluor 647 was excited at 633 nm. The optical pulse-chase assay was previously described in refs. 24,55. Briefly, post blocking, channels at the surface were first labeled with either α-bungarotoxin (Life Technologies) or biotin-conjugated α-bungarotoxin (Biotin-BTX; Life Technologies), before allowing for channel trafficking to resume for various times at 37 C, followed by labeling with 1 μM Alexa Fluor 647 conjugated α-bungarotoxin (BTX-647; Life Technologies) or 1 μM Alexa Fluor 647 conjugated Streptavidin (SA-647; Life Technologies) to measure channel forward and reverse trafficking, respectively.

## Electrophysiology

Patch clamp recordings of KCNQ1 or KCNQ1 + KCNE1 currents ($I_{Ks}$) were as previously described[55,56]. Cells plated in 3.5 cm culture dishes were mounted on the stage of an inverted microscope (OLYMPUS BH2-HLSH, Precision Micro Inc., Massapequa, NY). Whole-cell patch clamp currents were recorded at room temperature using an Axopatch 200B amplifier (Axon Instruments, Foster City, CA). Voltage-clamp pulse protocol consisted of a 500-ms step at a -70-mV holding potential a family of 2-s test pulse potentials (from -40 to +100 mV in 20-mV increments) and then a 2-s repolarizing pulses to -40 mV during which $I_{Ks}$ tail current was measured. External solution contained: 132 mM NaCl, 4.8 mM KCl, 2 mM CaCl2, 1.2 mM MgCl2, 10 mM HEPES and 5 mM glucose (pH adjusted to 7.4 with NaOH). Internal solution contained: 110 mM KCl, 5 mM ATP-K2, 11 mM EGTA, 10 mM HEPES, 1 mM CaCl2 and 1 mM MgCl2 (pH adjusted to 7.3 with KOH). Pipette series resistance was typically 1.5 - 3 MΩ when filled with internal solution. Currents were sampled at 10 kHz and filtered at 5 kHz. Traces were acquired at a repetition interval of 10 seconds.

## Confocal imaging

At 24 hr post-transfection, HEK cells were split onto fibronectin-coated 8-well glass bottom chamber slides (Ibidi GmbH). In some conditions, cells were treated overnight with 5 μM MG132 or 100 μM chloroquine to block the proteasome or lysosome, respectively. The next day, cells were treated with 100 μM cycloheximide for 1 h and fixed with 4% paraformaldehyde (PFA) for 10 mins at room temperature (RT). Cells were washed three times with PBS and permeabilized for 10 mins with 0.2% Triton X-100 in PBS with 0.2 M glycine and 3% BSA. The cells were washed thrice with PBS and incubated 30 mins in blocking buffer (PBS with 0.2 M glycine and 3% BSA), followed by overnight incubation with antibodies targeting subcellular organelles [Calnexin (Ab22595),

RCAS1 (D2B6N), Rab9A (D52G8), EEA1 (C45B10), LAMP2 (L0668) and PSMA2 (2455S)]. The cells were then washed thrice with PBS and incubated for 1 h with HRP- conjugated goat anti-rabbit secondary antibody (1:5000) and imaged with a Nikon A1RMP confocal microscope using a 60x oil-immersion objective.

Cardiomyocytes were plated onto 35-mm MatTek dishes (MatTek Corporation) and fixed with 4% formaldehyde for 10 min at RT. Cells were stained with BTX-647 as described above. Images were captured as Z-stacks on a Nikon A1RMP confocal microscope with a 40x oil-immersion objective and converted to maximum intensity projections for analysis using Fiji software.

### Immunoprecipitation, Western blot, and proteomics sample preparation

HEK293/CHO cells were washed once with Ca$^{2+}$-free PBS, harvested, and resuspended in RIPA lysis buffer containing (in mM): Tris (20, pH 7.4), EDTA (1), NaCl (150), 0.1% (wt/vol) SDS, 1% Triton X-100, 1% sodium deoxycholate and supplemented with protease inhibitor mixture (10 µL/mL, Sigma-Aldrich, St. Louis, MO), PMSF (1 mM, Sigma-Aldrich), N-ethylmaleimide (2 mM, Sigma-Aldrich) and PR-619 deubiquitinase inhibitor (50 µM, LifeSensors). Lysates were prepared by incubation at 4 °C for 1 hr, with occasional vortexing, and cleared by centrifugation (10,000 × g, 10 min, 4 °C). Supernatants were transferred to clean tubes, with aliquots removed for quantification of total protein concentration using the BCA protein estimation kit (Pierce Technologies).

For immunoprecipitation, lysates were pre-cleared by incubation with 20 µL Protein A/G Sepharose beads (Rockland) bound to anti-rabbit IgG (Sigma) for 3 h at 4 °C. Equivalent total protein amounts were added to spin-columns containing 50 µL Protein A/G Sepharose beads incubated with 3 µg anti-KCNQ1 (Alomone, Jerusalem, Israel), and tumbling overnight at 4 °C. Immunoprecipitates were washed 3–5 times with RIPA buffer, spun down at 500 × g, eluted with 40 µL of warmed sample buffer [50 mM Tris, 10% (vol/vol) glycerol, 2% SDS, 100 mM DTT, and 0.2 mg/mL bromophenol blue], and boiled (55 °C, 15 min). Proteins were resolved on a 4–12% Bis·Tris gradient precast gel (Life Technologies) in MOPS-SDS running buffer (Life Technologies) at 200 V constant for ~1 hr. We loaded 10 µL of the PageRuler Plus Prestained Protein Ladder (10–250 kDa, Thermo Fisher, Waltham, MA) alongside the samples.

For proteomic analysis, the gels were stained with SimplyBlue (ThermoFisher Scientific) and KCNQ1 monomer and dimer bands were excised. In-gel digestion was performed as previously described[57], with minor modifications. Gel slices were washed with 1:1 Acetonitrile and 100 mM ammonium bicarbonate for 30 min then dehydrated with 100% acetonitrile for 10 min until shrunk. The excess acetonitrile was removed, gel slices were dried in speed-vacuum at room temperature for 10 minutes and then reduced with 5 mM DTT for 30 min at 56 °C in an air thermostat, cooled down to room temperature, and alkylated with 11 mM IAA for 30 min with no light. Gel slices were then washed with 100 mM of ammonium bicarbonate and 100% acetonitrile for 10 min each. Excess acetonitrile was removed and dried in a speed-vacuum for 10 min at room temperature and the gel slices were re-hydrated in a solution of 25 ng/µL trypsin in 50 mM ammonium bicarbonate for 30 min on ice and digested overnight at 37 °C in an air thermostat. Digested peptides were collected and further extracted from gel slices in extraction buffer (1:2 ratio by volume of 5% formic acid: acetonitrile) at high speed, shaking in an air thermostat. The supernatants from both extractions were combined and dried in a speed-vacuum. Peptides were dissolved in 3% acetonitrile/0.1% formic acid.

For Western blotting, protein bands were transferred from the gel by tank transfer onto a nitrocellulose or Immobilon-P PVDF membrane (Sigma-Aldrich) (3.5 hr, 4 °C, 30 V constant) in transfer buffer (25 mM Tris pH 8.3, 192 mM glycine, 15% (vol/vol) methanol, and 0.1% SDS). The membranes were blocked with a solution of 5% nonfat milk (BioRad) in tris-buffered saline-tween (TBS-T) (25 mM Tris pH 7.4, 150 mM NaCl, and 0.1% Tween-20) for 1 hr at RT and then incubated overnight at 4 °C with one of the primary antibodies (anti-KCNQ1, Alomone, APC-022, 1:1000 or anti-actin antibody (1:2000, Abcam, USA) in blocking solution. The blots were washed with TBS-T three times for 10 min each and then incubated with secondary horseradish peroxidase-conjugated antibody for 1 hr at RT. After washing in TBS-T, the blots were developed with a chemiluminescent detection kit (Pierce Technologies) and then visualized on a gel imager. Membranes were then stripped with harsh stripping buffer (2% SDS, 62 mM Tris pH 6.8, 0.8% ß-mercaptoethanol) at 50 °C for 30 min, rinsed under running water for 2 min, and washed with TBST (3x, 10 min). Membranes were blocked with 5% milk again and re-blotted with anti-ubiquitin (LifeSensors, VU1, 1:500). The intensity of Ub smear in each lane was quantified and normalized to the sum of KCNQ1 monomer and dimer band intensities. The unprocessed raw images of the blots are included in the source file.

### Liquid chromatography with tandem mass spectrometry (LC-MS/MS)

Desalted peptides were injected in an EASY-SprayTM PepMapTM RSLC C18 50 cm X 75 cm ID column (Thermo Scientific) connected to an Orbitrap FusionTM TribridTM (Thermo Scientific). Peptides elution and separation were achieved at a non-linear flow rate of 250 nL/min using a gradient of 5%-30% of buffer B (0.1% (v/v) formic acid, 100% acetonitrile) for 110 minutes with a temperature of the column maintained at 50 °C during the entire experiment. The Thermo Scientific Orbitrap Fusion Tribrid mass spectrometer was used for peptide tandem mass spectroscopy (MS/MS). Survey scans of peptide precursors are performed from 350 to 1500 m/z at 120 K full width at half maximum (FWHM) resolution (at 200 m/z) with a $2 \times 10^5$ ion count target and a maximum injection time of 60 ms. The instrument was set to run in top speed mode with 3-second cycles for the survey and the MS/MS scans. After a survey scan, MS/MS was performed on the most abundant precursors, i.e., those exhibiting a charge state from 2 to 6 of greater than $5 \times 10^3$ intensity, by isolating them in the quadrupole at 1.6 Th. We used Higher-energy C-trap dissociation (HCD) with 30% collision energy and detected the resulting fragments with the rapid scan rate in the ion trap. The automatic gain control (AGC) target for MS/MS was set to $5 \times 10^4$ and the maximum injection time was limited to 30 ms. The dynamic exclusion was set to 30 s with a 10 ppm mass tolerance around the precursor and its isotopes. Monoisotopic precursor selection was enabled.

### Data and statistical analysis

Patch clamp data, shown as mean ± S.E., were acquired using pCLAMP 8.0 (Axon Instruments) and analyzed with Origin 7.0 (OriginLab, Northampton, MA) and Clampfit 8.2 (Axon Instruments). Flow cytometry data were analyzed using FlowJo 10.8 software. Raw mass spectrometric data were analyzed using the Proteome Discoverer 2.4 to perform database search and LFQ quantification at default settings. PD2.4 was set up to search with the reference human proteome database downloaded from UniProt and performed the search trypsin digestion with up to 2 missed cleavages. Peptide and protein false discovery rates (FDR) were all set to 1%. The following modifications were used for protein identification and LFQ quantification: Carbamidomethyl(C) was set as fixed modification and variable modifications of Oxidation (M) and Acetyl (Protein N-term), DiGly (K) and deamination for asparagine or glutamine (NQ). Results obtained from PD2.4 were further used to quantify relative diGly peptide abundances under the different conditions and identifying sites modified on KCNQ1. Statistical data analysis was assessed with Student's t-test for comparison between two groups, one-way ANOVA for comparisons among more than two groups, and two-way ANOVA for more than one variable per group. In the Figures, statistically significant differences from

control values are marked by asterisks (*, $p < 0.05$; **, $p < 0.01$; ***, $p < 0.001$; #, $p < 0.0001$), and the *P* values with 95% confidence intervals are reported in the respective figure legends.

## Material availability

Plasmid constructs for non-commercial purposes can be obtained by request from the corresponding author after publication of the manuscript.

## Reporting summary

Further information on research design is available in the Nature Portfolio Reporting Summary linked to this article.

## Data availability

The authors declare that all data supporting the findings of this study are available within the paper and its supplementary information files. Source data are provided with this paper. The mass spectrometry proteomics data have been deposited to the ProteomeXchange Consortium via the PRIDE partner repository with the dataset identifier PXD057805. Project Name: Digly analysis of human Kv7.1 transfected in HEK293 cells. Source data are provided with this paper.

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

## Acknowledgements

We thank Ming Chen for technical support, and Drs. Arden Darko-Boateng and Yuki Utsugi for comments on the manuscript. This work was supported by grants RO1-HL121253, R01-HL142111, R01-NS126850 and P01-HL164319 from the NIH (to HMC); AHA postdoctoral fellowship POST1019343 (to SKS); Medical Scientist Training Program grant (T32 GM007367) and NHLBI National Research Service Award (1F30-HL140878) (to SAK); EA was supported by NIGMS postdoctoral training grant T32 GM008464-32. Flow cytometry experiments were performed in the CCTI Flow Cytometry Core, supported in part by the NIH (S10RR027050). Confocal images were collected in the HICCC Confocal and Specialized Microscopy Shared Resource, supported by NIH (P30 CA013696).

## Author contributions

S.K.S. and S.A.K. designed and conducted experiments and analyzed data; X.Z., E.A., P.C., and R.S. conducted experiments and analyzed data; R.S.K. designed experiments and edited the paper; H.M.C. designed experiments and obtained funding; S.K.S. and H.M.C. wrote the paper.

## Competing interests

S.A.K. and H.M.C. are inventors on a patent held by Columbia University for "Compositions and methods for using engineered deubiquitinases for probing ubiquitin-dependent cellular processes." H.M.C. is a scientific co-founder and on the SAB of two startups, Stablix, Inc. and Flux Therapeutics, pursuing targeted protein stabilization therapeutics. S.A.K. is a co-founder and employee of Stablix, Inc. The remaining authors declare no competing interests.
