## [Transparent Peer Review file · Nature Communications]

Decoding polyubiquitin regulation of KV7. 1 (KCNQ1) surface expression with engineered linkage-selective deubiquitinases

Corresponding Author: Dr Henry Colecraft

Version 0:

Reviewer comments:

Reviewer #1

(Remarks to the Author)

Colecraft and colleagues used linkage-selective catalytic domains of deubiquitinases (DUBs) fused to GFP-targeted nanobody to study polyubiquitin linkage regulation of the voltage-gated potassium channel KCNQ1. Their key findings include that K11 linkages promote ER retention/degradation, enhance endocytosis, and reduce recycling of the channel; K29/K33 linkages promote ER retention/degradation; K63 linkages enhance endocytosis and reduce recycling; and K48 linkages are necessary for forward trafficking.

Overall, the authors generated a broadly applicable tool to investigate the functional role of ubiquitin linkage types in diverse polyubiquitin signals in cellulo, which has been challenging so far. Using this tool, they revealed unknown roles for K11 and K29/33 ubiquitin linkages in controlling the subcellular localization and trafficking of KCNQ1. They also showed that the ubiquitin code of KCNQ1 is cell-context-dependent.

Publication could be considered after addressing the following comments:

Figure 1:

B: Why have only 8-10 cells been analyzed (also in Fig. 2F-J, S1C)? A minimum of 20-30 cells per condition and three biological replicates should be analyzed to account for biological variability and ensure reliable results.

I: Although the ubiquitin signal upon KCNQ1-YFP IP is reduced in the presence of enDUBs, no proof was included that enDUBs are equally expressed in the cells (input blot is missing)

J: Which anti-KCNQ1 signal was quantified for the analysis? Only the monomer band or the entire lane?

Figure 2:

B-E: How did the authors ensure that the YFP-positive cells actually express the enDUBs (equally strong)? Did they first gate for CFP-positive cells?

In general, no figure is included that shows the enDUB expression and actual binding to KCNQ1-YFP.

Figure 7:

C/D: How has the SEM in these graphs been calculated? Based on $SEM = SD \text{ divided by the square root of } N$ (and assuming that $N=3$), the values are far too small.

F-J: Do these graphs show one of the two biological replicates? Meaning, the one-way ANOVA test was done taking only one replicate into account?

In general: What is the consequence of enDUB expression on the total cellular conjugated ubiquitin pool? Include one Western blot showing an anti-ubiquitin staining of a total cell lysate expressing the enDUBs alone and in combination with KCNQ1-YFP.

Minor comments:

Title: Why has Kv7.1 been used for the title and KCNQ1 throughout the text?

Page 9: When describing results of Figure 7, the authors refer to Figure 6.

Page 10: When mentioning LQT1 it would be helpful to define the abbreviation/ briefly describe the syndrome.
Fig. 1A: Include the protein names that were stained as organelle markers in the figure legend or figure (also in Fig. S7).
Fig. 1D: The graphs would be more intuitive if treatment conditions were indicated.
Fig. 4: I suggest to call the E3 ligase always the same (NEDD4-2 vs. NEDD4L)
Fig. 4G: I suggest to choose another color scheme for the pie charts (the ubiquitin linkage distribution). In Fig. 4A-F, black and red has been used for nano and NEDD4-2, therefore it was a bit confusing.
Fig. S5: What are the asterisks indicating (I assume catalytic inactivity)?

Reviewer #2

(Remarks to the Author)

In this manuscript, a collection of linkage-specific engineered deubiquitinases (enDUBs) that selectively hydrolyze linkages from a given cellular target were developed to exploit polyubiquitin linkage preferences. The authors investigated this in the context of the cellular posttranscriptional regulation of KCNQ1, an ion channel that is important for an array of cellular processes. The authors claim that their results reveal distinct behavior of the enDUBs, which supposedly reflects the differential regulation of KCNQ1 trafficking by different polyubiquitin chain.

The methodology to address polyubiquitin regulation makes use of linkage-selective enDUBs by fusing linkage-selective DUB catalytic domains to GFP-targeted nanobody. The fate of the target protein YFP-KCNQ1 was then studied in various cell types. As the readout technology, the authors use a strong combination of mass spectrometry and functional (patch-clamp) studies.

Overall, the study shows interesting, though often somewhat surprising results, which are difficult to place in the context of the current knowledge of posttranscriptional polyubiquitin regulation.

One major concern I have is that both the enDUBs and the ion channel KCNQ1 were transiently transfected into the cells. Therefore, the expression levels may not reflect endogenous expression levels of both target protein and DUBs. The posttranslational polyubiquitin linkage regulation of target proteins is a delicate process that may be influenced by imbalances in the proteome. This makes it a rather artificial experimental system to study these sensitive posttranscriptional processes.

Another major concern is how specific the linkage-selective enDUBs used in this work really are and how the specificity could be compromised by prolonged tethering of DUB to substrate enabled by the nanobody. In the Discussion section, the authors admit that this could jeopardize the results obtained in this study. Some findings are indeed rather surprising (e.g., KCNQ1-YFP expression was upregulated by enDUB-Cz, enDUB-Tr, and enDUB-U21, but not enDUB-O1 or enDUB-O4, indicating a dominant contribution of K11 and K29/K33 but not K63 or K48 to KCNQ1-YFP degradation) and although the authors come up with alternative explanations, it is difficult to judge whether these findings represent the endogenous cellular situation.

I have some other questions and comments, listed below in bullet points.

- The authors focus on the regulation of the ion channel KCNQ1, but do not explain why they have chosen this target protein. Was there a particular reason for choosing a membrane protein for these studies? Why didn't the authors take a selection of various proteins (membrane associated, cytosolic, etc.)?

- P5: MG132 inhibits the proteasome and thus hinders the recycling of polyubiquitin. What is the effect of MG132 on the prevalence and availability of free monomeric ubiquitin or free (non-conjugated) polymeric ubiquitin in the cell and the biosynthesis of new ubiquitin? And how could this affect polyubiquitin linkage regulation?

- P5: I have concerns about the term 'expression' used throughout the manuscript, such as in "[...] distinct polyubiquitin linkage chains could represent a prominent mechanism for controlling KCNQ1-YFP expression [...]", "[...] Ubiquitination is an intricate posttranslational modification that regulates the functional expression of all proteins to control biology [...]", etc. (Poly)ubiquitination controls degradation, trafficking, etc., i.e. cellular processes that take place at the posttranslational level and - thus - influence cellular protein abundance and localization.

- P5, Fig 1E: There are a lot of background proteins present in the IP fraction. How can the authors be sure that the ubiquitin (diGly-)peptides they identify in the excised gel band is from (poly)ubiquitin that is conjugated to KCNQ1 and not to other proteins? A more stringently washed IP fraction could have given a cleaner KCNQ1 isolation and less uncertainty about this. Also, longer polyubiquitin chains may be missed if the band was sliced out too narrowly. I would suggest to perform the IP under more stringent conditions and omit the SDS-PAGE step to do on-bead digestions, to make sure that all KCNQ1 (including all polyubiquitinated species) is captured.

- Fig 11: OTUD-4 is specific for K48 linkage, which is responsible for ~75% of all channel ubiquitination. As a consequence, one would expect enDUB-O4 to have the most prominent effect in this assay. This is not what is observed though: the effect seems to be more or less the same for all enDUBs (except for enDUB-U21): can the authors comment on this?

- P5: the authors hypothesize that the unexpected results presented in Fig 11 may be the results of overexpression of the enDUBs (vide supra for my major concern). Have they done titration experiments to study the effect of decreased expression

levels of the enDUBs, that would approach endogenous levels of linkage specific DUBs?

- P5: the authors find that EnDUB-O4 (K48) had no effect on KCNQ1 total abundance, but that it unexpectedly reduced channel surface density. This is indeed surprising, as K48 is the consensus proteasomal degradation signal. Can the authors comment on this and provide alternative explanations? In this respect, I miss data on the global KCNQ1 levels in these cells, which could give information about the degradation of the channel. Global protein levels could be measured by global proteomics, have the authors considered performing such experiments?

- P8, Fig 5B: In the blot for ubiquitin on the right, wouldn't the authors expect a differential staining pattern for each of the enDUBs? It now seems that all ubiquitination is gone from KCNQ1 in all cases, which is not what one would expect if enDUBs were specific. In contrast, in Fig 6B one would expect the effect of enDUB-O4 to be the highest, as the K48 linkage is generally responsible for most of the ubiquitin signal. Could the authors comment on that?

- P13, Discussion: "[...] distinct KCNQ1 LQT1 mutations were differentially rescued by linkage-selective enDUBs. Surface density of R366W and R539W were most robustly increased by enDUB-Tr while T587M and G589D were most responsive to enDUB-O1 [...]". How can ubiquitination patterns be different in the mentioned (non-Lysine) mutants? Is non-canonical ubiquitination at different amino acid residues involved here? Or do conformational changes that prevent Lys ubiquitination play a role?

Reviewer #3

(Remarks to the Author)

The authors design target-selective, ubiquitin (Ub)-linkage-preferred deubiquitinases 'DUBs' to deduce whether and how ubiquitylation of KCNQ1 (the target) with different configurations of polyUb chains can influence the expression, trafficking/distribution and function of this ion channel. The 'Big Picture' questions the authors want to address are: (1) Is it possible to explore the functional consequences of ubiquitylation with different polyUb chain configurations (the Ub codes) in living cell experiments? (2) Do the functional consequences of Ub code usage depend on the cell type, cellular context, and protein sequences?

To achieve their goals, the authors create two sets of molecular tools. First, they create a KCNQ1 construct (BBS-Q1-YFP) that allows fluorescent labeling of cell surface KCNQ1 (fluorescent batrachotoxin that binds to the extracellularly oriented BTX binding sequence fused into KCNQ1), with YFP fused to the cytoplasmic N-terminus that can be used as a read out of total KCNQ1 as well as the tether site of GFP/YFP-nanobody. Second, they fuse the catalytic domains of 4 Ub-linkage preferred DUBs and one nonselective DUB to the GFP/YFP-nanobody (called engineered or enDUBs). The authors validate these molecular tools by showing that when BBS-Q1-YFP and enDUBs are coexpressed in cells, the degree of Q1 ubiquitylation is drastically reduced. A KCNQ1 construct without YFP or a catalytically dead enDUB does not have this effect.

With these molecular tools, the authors can evaluate the effects of manipulating KCNQ1 ubiquitylation on the total expression, cell surface expression and subcellular distribution of KCNQ1 using high-throughput fluorescence flow cytometry, confocal microscopy, and patch clamp recording. They also use proteomics to explore the pattern of Ub-linkages on KCNQ1 as well as the potential ubiquitylation sites on KCNQ1. These experiments largely answer YES to question (1) listed above.

In terms of question (2), the authors compare the effects of enDUBs on the total and surface expression of KCNQ1 between HEK293 cells and guinea pig ventricular myocytes (cell-type dependence?), between basal conditions vs when KCNQ1 ubiquitylation is markedly elevated by coexpressing KCNQ1 E3 ubiquitin ligase, NEDD4-2 or ITCH (cellular context dependence?), and between WT KCNQ1 vs trafficking-defective KCNQ1 mutants (protein sequence dependence?). These experiments answer YES to question (2).

Overall, the authors provide novel results that are of significance to the field of protein ubiquitylation. Ubiquitylation is a ubiquitous system of protein post-translational modifications, that influence proteins' degradation, trafficking/distribution and thus functions. Ubiquitin (Ub) can be covalently linked to proteins' lysine side chains as monoUb or multi-monoUb, or as straight or branched polyUb chains via linkage through their own 7 Lys side chains as well as the Met1 -NH₂ group. The latter group of straight or branched polyUb chains created limitless configurations of Ub linkage. Our understanding of the complexity of protein ubiquitylation has been greatly aided by the recent development of antibodies, aptamers and affimers that can recognize different Ub-linkages, allowing global pull down followed by proteomics analysis. The current study focuses on decoding Ub-linkages in living cells, that complements the aforementioned Ab(-like)-based/proteomics approaches.

Although this study has many strengths listed above, it would benefit from some additional data and clarifications of some experimental procedures:

1. Immunostaining of organelles: Based on Fig. 1D and Fig. S2, the authors concluded that KCNQ1 is mainly degraded by proteasomes, instead of lysosomes. I would like to suggest that the authors add proteasome immunostaining to Fig. 1A and Fig. 2F-J, along with quantification of KCNQ1-proteasome colocalization.
2. Fig. 7 depicts the images of guinea pig ventricular myocytes expressing BBS-Q1-YFP and nano or different enDUBs. The data allowed the authors to quantify total and surface expression of Q1, as well as the expression level of enDUBs. Because ventricular myocytes are 3D structures, the results of image quantification will depend on the z-planes of confocal imaging. Please comment on the following: (a) How did you choose the z-plane of confocal imaging and the reason behind the choice? (b) Why the YFP and 647 images of enDUB-O1/-Cz and -U21 show such distinct patches? What could be the subcellular compartments of these patches? (c) Have you run 3D imaging of the whole myocyte thickness, and were the results similar to, or different from, the data presented in Fig. 7C and Fig. 7D? (d) It will be of great value to monitor the distribution of secretory- and endocytic-pathway organelles in myocytes. Since you have used green, red, and far-red

channels, the blue channel can be used for the organelles.

3. Fig. 1I and 1J are critical in showing the effectiveness of the catalytic domains of enDUBs. Please clarify the following: (a) What is the nature of the 100 kDa band in both Q1 and Ub IB of the UT (untransfected? not explained in figure legend) lane? (b) If the 100 kDa band in Q1-transfected lanes represents Q1 monomer, why its Ub band intensity is not reduced or even increased in enDUB coexpressed lanes vs the 'nano' lane? (c) Although the Q1 IB image shows distinct monomer, dimer, trimer and tetramer bands, the Ub IB shows a smear. What may cause this smear in Ub IB? Did you quantify Q1 ubiquitylation shown in Fig. 1J based on the distinct Q1 bands or based on the smear of the whole lanes? (d) The Ub IB image shows alarming unevenness in the 'nano' lane, as well as in the following 3 enDUB lanes. This problem calls into question the validity of Fig. 1J. Please consider replacing the Ub IB image with a better 'representative' one.
4. Fig. 3F: The differences among the BTX-647 decay time courses are not clear, except the cyan curve. Would a semi-log plot be more effective?
5. Some of the patch clamp experiments were done on coexpressed KCNQ1/KCNE1, while others on KCNQ1 alone. Why this difference? KCNE1 interacts with KCNQ1's C-terminus where the putative Ub sites are located. Does KCNE1 affect KCNQ1 ubiquitylation?
6. Three minor points: (a) NEDD4-2 or NEDD4L: Both terms were used in the text. Please clarify/unify. (b) Fig. 1E-1G: need info on how the UT negative control was used in quantification of mass spectrometry data. (c) Fig. S1, the Q1-YFP label is misplaced.

Version 1:

Reviewer comments:

Reviewer #1

(Remarks to the Author)

The authors have sufficiently addressed all my comments. I can recommend publication of the manuscript in its current form.

Reviewer #2

(Remarks to the Author)

I think that the authors have adequately addressed the comments made by the reviewers in the revised version of the manuscript. Therefore, I have no further comments.

Reviewer #3

(Remarks to the Author)

My concerns have been properly addressed by the authors.

Point -to-Point Response to Reviewer comments

We thank the reviewers for their positive and constructive feedback on this manuscript. We have revised the manuscript to address the concerns raised by the reviewers. Please find below the point-by-point response addressing the reviewers' comments.

REVIEWER COMMENTS

Reviewer #1:

Colecraft and colleagues used linkage-selective catalytic domains of deubiquitinases (DUBs) fused to GFP-targeted nanobody to study polyubiquitin linkage regulation of the voltage-gated potassium channel KCNQ1. Their key findings include that K11 linkages promote ER retention/degradation, enhance endocytosis, and reduce recycling of the channel; K29/K33 linkages promote ER retention/degradation; K63 linkages enhance endocytosis and reduce recycling; and K48 linkages are necessary for forward trafficking.

Overall, the authors generated a broadly applicable tool to investigate the functional role of ubiquitin linkage types in diverse polyubiquitin signals in cellulo, which has been challenging so far. Using this tool, they revealed unknown roles for K11 and K29/33 ubiquitin linkages in controlling the subcellular localization and trafficking of KCNQ1. They also showed that the ubiquitin code of KCNQ1 is cell-context-dependent.

Publication could be considered after addressing the following comments:

Figure1:

B: Why have only 8-10 cells been analyzed (also in Fig. 2F-J, S1C)? A minimum of 20-30 cells per condition and three biological replicates should be analyzed to account for biological variability and ensure reliable results.

We have expanded the confocal analysis to include at least 20 cells per condition across three independent biological replicates. The additional data are included in Figures 1B, 2F-J, and Supplemental Figures S1 and S9.

I: Although the ubiquitin signal upon KCNQ1-YFP IP is reduced in the presence of enDUBs, no proof was included that enDUBs are equally expressed in the cells (input blot is missing).

The enDUBs are expressed in a P2A-CFP vector enabling confirmation of their expression by either monitoring CFP fluorescence or by using anti-GFP antibodies in Western blots. For flow cytometry experiments, we gated cells for YFP- and CFP-expression, thus ensuring the co-expression of both KCNQ1-YFP and enDUBs in analyzed cells. We have now included quantification of CFP fluorescence from flow cytometry experiments as well as an input anti-GFP (CFP) blot from a Western blot experiment to indicate relatively equitable expression of the different enDUBs

(Supplemental Fig. 5). Importantly, the qualitative functional differences amongst the enDUBs cannot be explained by potential differences in expression levels.

J: Which anti-KCNQ1 signal was quantified for the analysis? Only the monomer band or the entire lane?

The monomeric and dimeric KCNQ1 bands were the most consistent band in different Western blot trials. We, therefore, used the monomer and dimer bands for KCNQ1 quantification to maximize consistency. On some occasions in pulldown experiments, we observed a non-specific band (potentially attributable to a dimer of the heavy chain of the antibody used for pulldown) in the untransfected lane that ran at a molecular weight (~100 kDa) similar to that of the KCNQ1 monomer. On these occasions, we subtracted the integrated signal of the nonspecific band in the untransfected lane from the corresponding signals in the experimental condition lanes. This information has now been added in the figure legend and in the methods section.

Figure 2:B-E: How did the authors ensure that the YFP-positive cells actually express the enDUBs (equally strong)? Did they first gate for CFP-positive cells? In general, no figure is included that shows the enDUB expression and actual binding to KCNQ1-YFP.

We gated on CFP and YFP positive cells for flow cytometry analysis to ensure that the measured population of cells express both BBS-Q1-YFP and the different enDUBs expressed in P2A-CFP plasmids. We included a negative control showing that the enDUBs had no impact on surface density of untagged KCNQ1 (Supplemental Fig. 8) demonstrating that binding to YFP is necessary for the observed effects.

Figure 7:

C/D: How has the SEM in these graphs been calculated? Based on $SEM = SD$ divided by the square root of N (and assuming that $N=3$), the values are far too small.

F-J: Do these graphs show one of the two biological replicates? Meaning, the one-way ANOVA test was done taking only one replicate into account?

In Fig 7, the YFP and 647 fluorescence intensities of each cell collected across three independent biological replicates using the same excitation intensity and gain settings on the confocal microscope. The intensity measurements for individual cells from all three trials are shown in the figure, along with the mean fluorescence intensities. The SEM was computed by dividing the SD by the square root of the total number of cells from all three experiments.

In general: What is the consequence of enDUB expression on the total cellular conjugated ubiquitin pool? Include one Western blot showing an anti-ubiquitin staining of a total cell lysate expressing the enDUBs alone and in combination with KCNQ1-YFP.

We performed Western blot analysis to assess the impact of enDUBs on total cellular ubiquitin. We observed no significant change in ubiquitin signals in total lysate expressing enDUBs alone or in combination with KCNQ1-YFP (please see figures below). The results are consistent with the negative control data shown in Supplemental Fig. 8 that the enDUBs used here are specific to YFP tagged proteins.

Minor comments:

Title: Why has Kv7.1 been used for the title and KCNQ1 throughout the text?

We have revised the title to include KCNQ1.

Page 9: When describing results of Figure 7, the authors refer to Figure 6.

We thank the reviewer for catching this error, we corrected this in the Results section.

Page 10: When mentioning LQT1 it would be helpful to define the abbreviation/ briefly describe the syndrome.

We have inserted a new paragraph in the Introduction that expands on the background to ion channels, trafficking defects and ion channelopathies. We define LQT1 and describe the syndrome at first mention in the new paragraph.

Fig. 1A: Include the protein names that were stained as organelle markers in the figure legend or figure (also in Fig. S7).

We have now included the names of the proteins that were stained as organelle markers in the relevant Figure legends (Figs. 1A, S1 and S9).

Fig. 1D: The graphs would be more intuitive if treatment conditions were indicated.

We have added labels indicating treatment conditions to the CDF plots in Fig 1D.

Fig. 4: I suggest to call the E3 ligase always the same (NEDD4-2 vs. NEDD4L)

Thank you. We have now maintained NEDD4-2 throughout the revised manuscript for consistency.

Fig. 4G: I suggest to choose another color scheme for the pie charts (the ubiquitin linkage distribution). In Fig. 4A-F, black and red has been used for nano and NEDD4-2, therefore it was a bit confusing.

Thank you. Done.

Fig. S5: What are the asterisks indicating (I assume catalytic inactivity)?

The asterisks indicate catalytically dead enDUBs. We have now included this description in the supplemental Figure S6 legend.

Reviewer #2:

In this manuscript, a collection of linkage-specific engineered deubiquitinases (enDUBs) that selectively hydrolyze linkages from a given cellular target were developed to exploit polyubiquitin linkage preferences. The authors investigated this in the context of the cellular posttranscriptional regulation of KCNQ1, an ion channel that is important for an array of cellular processes. The authors claim that their results reveal distinct behavior of the enDUBs, which supposedly reflects the differential regulation of KCNQ1 trafficking by different polyubiquitin chain.

The methodology to address polyubiquitin regulation makes use of linkage-selective enDUBs by fusing linkage-selective DUB catalytic domains to GFP-targeted nanobody. The fate of the target protein YFP-KCNQ1 was then studied in various cell types. As the readout technology, the authors use a strong combination of mass spectrometry and functional (patch-clamp) studies.

Overall, the study shows interesting, though often somewhat surprising results, which are difficult to place in the context of the current knowledge of posttranscriptional polyubiquitin regulation.

One major concern I have is that both the enDUBs and the ion channel KCNQ1 were transiently transfected into the cells. Therefore, the expression levels may not reflect endogenous expression levels of both target protein and DUBs. The posttranslational polyubiquitin linkage regulation of target proteins is a delicate process that may be influenced by imbalances in the proteome. This makes it a rather artificial experimental system to study these sensitive posttranscriptional processes.

We agree with the reviewer that the over-expression experimental paradigm is a limitation to this study which we now indicate in the discussion section. Nevertheless, the experimental system also has some unique advantages that we exploit to validate linkage-selective enDUBs as a prototype technology that can be applied to address longstanding in the questions in the ubiquitin field pertaining to the functional relevance of distinct polyubiquitin linkages on specific proteins in live cells. We have tried to provide a balanced consideration of the advantages and limitations of the study in the Discussion. Now that we have validated the technology, we are applying it to probe polyubiquitin regulation of endogenous proteins in new studies that are beyond the scope of this work.

Another major concern is how specific the linkage-selective enDUBs used in this work really are and how the specificity could be compromised by prolonged tethering of DUB to substrate enabled by the nanobody. In the Discussion section, the authors admit that this could jeopardize the results obtained in this study. Some findings are indeed rather surprising (e.g., KCNQ1-YFP expression was upregulated by enDUB-Cz, enDUB-Tr, and enDUB-U21, but not enDUB-O1 or enDUB-O4, indicating a dominant contribution of K11 and K29/K33 but not K63 or K48 to KCNQ1-YFP degradation) and although the authors come up with alternative explanations, it is difficult to judge whether these findings represent the endogenous cellular situation.

The interpretation of our results hinges critically on previous work showing that the DUBs selected for this study have remarkable specificity for hydrolyzing distinct linkage types *in vitro*. The structural basis for this specificity has also been demonstrated for some DUBs. The qualitative differences we observe of the distinct linkage-specific enDUBs effects on KCNQ1 abundance and subcellular localization is the best evidence that the cells can distinguish their distinctive polyubiquitin chain hydrolysis signatures on this particular substrate, despite their prolonged tethering.

One way to increase confidence that the observed effects and interpretation are correct would be to determine whether different DUBs that have the same polyubiquitin linkage specificity would give the same functional outcome on a substrate when they are incorporated into linkage-selective enDUBs. To this end, we tested another K48-specific DUB, OTUB1, to see whether it recapitulated the unexpected impact of OTUD4 in decreasing the surface density of KCNQ1-YFP. Reassuringly, enDUB-OTUB1 mimicked enDUB-OTUD4 in suppressing KCNQ1 surface density without impacting channel abundance. Thus, this provides independent corroboration of the provocative

result that in this context, K48 linkages of KCNQ1 are important for channel forward trafficking but not protein stability. The new results are included in Supplemental Fig. S7.

I have some other questions and comments, listed below in bullet points.

- The authors focus on the regulation of the ion channel KCNQ1, but do not explain why they have chosen this target protein. Was there a particular reason for choosing a membrane protein for these studies? Why didn't the authors take a selection of various proteins (membrane associated, cytosolic, etc.)?

A motivation for our study was to develop an approach that enabled interrogation of hypothesized distinct functions of diverse polyubiquitin chains on individual proteins in live cells – a technology that does not exist as we indicate, in the Introduction and Discussion. The choice of KCNQ1 was chosen for several reasons: 1) it is an important protein involved in important physiological functions in heart, auditory system and beyond; 2) mutations in the protein cause life-threatening cardiac arrhythmias and deafness; 3) it is convenient because there are diverse outcomes we can readily measure— e.g. abundance, subcellular localization, and functional currents— to provide a readout of distinctive effects of linkage-selective enDUBs; 4) our lab has been focused in ion channel regulation and we have a lot of experience with this channel. We believe that for these reasons, the choice of KCNQ1 as a model protein to develop and validate the linkage-selective enDUB technology is reasonable. With this validation, the approach can now be applied to other proteins which will be one important focus of the lab moving forward.

We have inserted a new paragraph in the Introduction that expands on the background to ion channels, trafficking defects and ion channelopathies. We define LQT1 and describe the syndrome at first mention in the new paragraph

- P5: MG132 inhibits the proteasome and thus hinders the recycling of polyubiquitin. What is the effect of MG132 on the prevalence and availability of free monomeric ubiquitin or free (non-conjugated) polymeric ubiquitin in the cell and the biosynthesis of new ubiquitin? And how could this affect polyubiquitin linkage regulation?

Treatment with MG132 for as little as 20 mins is reported to be sufficient to observe significantly less free ubiquitin in the cell and leads to an increase in ubiquitinated protein species (Patrick et al., *Currents Biology*, 2003). In our experiments, we use MG132 in limited experiments to measure the consequence of inhibiting the proteasome on KCNQ1 abundance and subcellular localization.

- P5: I have concerns about the term 'expression' used throughout the manuscript, such as in "[...] distinct polyubiquitin linkage chains could represent a prominent mechanism for controlling KCNQ1-YFP expression [...]", "[...] Ubiquitination is an intricate posttranslational modification that regulates the functional expression of all proteins to control biology [...]", etc. (Poly)ubiquitination controls degradation, trafficking, etc., i.e. cellular processes that take place at the posttranslational level and - thus - influence cellular protein abundance and localization.

We appreciate the reviewer's point here and agree that terminology regarding 'expression' and 'trafficking' can be made more precise. We have followed the reviewer's recommendation and replaced 'expression' and 'trafficking' with 'abundance' and 'subcellular localization' throughout the revised manuscript.

- P5, Fig 1E: There are a lot of background proteins present in the IP fraction. How can the authors be sure that the ubiquitin (diGly-)peptides they identify in the excised gel band is from (poly)ubiquitin that is conjugated to KCNQ1 and not to other proteins? A more stringently washed IP fraction could have given a cleaner KCNQ1 isolation and less uncertainty about this. Also, longer polyubiquitin chains may be missed if the band was sliced out too narrowly. I would suggest to perform the IP under more stringent conditions and omit the SDS-PAGE step to do on-bead digestions, to make sure that all KCNQ1 (including all polyubiquitinated species) is captured.

In mass spectrometry experiments, we utilized values from an untransfected control to subtract any non-specific ubiquitin peptides that may be present in the different conditions. We thank the reviewer for helpful suggestions to improve this methodology which we will adopt for future studies, including on native proteins.

- Fig 1I: OTUD-4 is specific for K48 linkage, which is responsible for ~75% of all channel ubiquitination. As a consequence, one would expect enDUB-O4 to have the most prominent effect in this assay. This is not what is observed though: the effect seems to be more or less the same for all enDUBs (except for enDUB-U21): can the authors comment on this?

We provided three possibilities to explain the unexpected finding that all the putative linkage-selective enDUBs were quite effective in diminishing the amount of ubiquitin pulled down with KCNQ1. The totality of the evidence is consistent with the presence of substantial heterotypic branched ubiquitin chains on the channel. It is worth emphasizing that the most compelling evidence that the distinct linkage-selective enDUBs have different effects is that the cell recognizes their activities as such and leads to qualitatively different outcomes with respect to their abundance and localization. Further, the Western blots and proteomics assays measure the global ubiquitin on all the channels expressed in the cell. The fraction of channels on the cell surface or in smaller organelles are a small fraction of the total channels which are in the ER.

Therefore, relying on the global ubiquitin signature as a proxy to explain the functional effects observed with the linkage-selective enDUBs would likely be confounding and lead to misinterpretation of the data.

- P5: the authors hypothesize that the unexpected results presented in Fig 11 may be the results of overexpression of the enDUBs (vide supra for my major concern). Have they done titration experiments to study the effect of decreased expression levels of the enDUBs, that would approach endogenous levels of linkage specific DUBs?

The response here is the same as to the preceding point. We have not done titration experiments in this work, but these would be important considerations for future studies focused on native proteins.

- P5: the authors find that EnDUB-O4 (K48) had no effect on KCNQ1 total abundance, but that it unexpectedly reduced channel surface density. This is indeed surprising, as K48 is the consensus proteasomal degradation signal. Can the authors comment on this and provide alternative explanations? In this respect, I miss data on the global KCNQ1 levels in these cells, which could give information about the degradation of the channel. Global protein levels could be measured by global proteomics, have the authors considered performing such experiments?

We have discussed the surprising result and implications of enDUB-O4 having no effect on KCNQ1 protein abundance and reducing surface density in the manuscript. We have also corroborated the result by finding that a different K48-linkage specific DUB, OTUB1, the same effect as enDUB-O4 when it is converted into an enDUB (enDUB-OTUB1). This is now included as Supplemental Fig. S7. The relevant sentences discussing implications of this result in the Discussion are:

"In HEK293 cells, KCNQ1-YFP expression was upregulated by enDUB-Cz, enDUB-Tr, and enDUB-U21, but not enDUB-O1 or enDUB-O4. These results indicate a dominant contribution of K11 and K29/K33 but not K63 or K48 to KCNQ1-YFP degradation in this cellular context. This result was unexpected because K48 was the most abundant linkage type associated with KCNQ1 and is the canonical polyubiquitin signal associated with protein degradation^{1,5,30,31}. Precedence for K48 polyubiquitination not resulting in protein degradation is provided by the yeast transcription factor, Met4, which was found to encode a ubiquitin interaction motif (UIM) that intramolecularly binds K48 polyubiquitin and protects the protein from proteasomal degradation³². Our results give reason to wonder whether KCNQ1 may similarly encode a UIM that protects the channel from K48 polyubiquitin-mediated proteasomal degradation. Future use of linkage-selective enDUBs may uncover that this phenomenon of non-degradative roles for K48 chains is more prevalent than currently appreciated."

- P8, Fig 5B: In the blot for ubiquitin on the right, wouldn't the authors expect a differential

staining pattern for each of the enDUBs? It now seems that all ubiquitination is gone from KCNQ1 in all cases, which is not what one would expect if enDUBs were specific. In contrast, in Fig 6B one would expect the effect of enDUB-O4 to be the highest, as the K48 linkage is generally responsible for most of the ubiquitin signal. Could the authors comment on that?

Please see responses above (-P5; Fig. 1I) which addresses the same point.

- P13, Discussion: “[...] distinct KCNQ1 LQT1 mutations were differentially rescued by linkage-selective enDUBs. Surface density of R366W and R539W were most robustly increased by enDUB-Tr while T587M and G589D were most responsive to enDUB-O1 [..]”. How can ubiquitination patterns be different in the mentioned (non-Lysine) mutants? Is non-canonical ubiquitination at different amino acid residues involved here? Or do conformational changes that prevent Lys ubiquitination play a role?

The reviewer raises some interesting questions and potential possibilities here. We do not know the basis for the differential putative ubiquitin patterns for different KCNQ1 mutations; this would be an interesting area for future studies. We speculate that different mutants may have altered folding/protein-protein interactions that contribute to changes in ubiquitination. There is precedent in cystic fibrosis transmembrane regulator (CFTR), for example, for a non-lysine mutation leading to altered ubiquitination of an ion channel. We refer to some previous work in the Discussion:

“These results agree with previous findings that mutations in ion channels can alter their ubiquitination status in a manner that interferes with their surface expression ⁴⁰⁻⁴³.”

Reviewer #3 (Remarks to the Author):

The authors design target-selective, ubiquitin (Ub)-linkage-preferred deubiquitinases 'DUBs' to deduce whether and how ubiquitylation of KCNQ1 (the target) with different configurations of polyUb chains can influence the expression, trafficking/distribution and function of this ion channel. The 'Big Picture' questions the authors want to address are: (1) Is it possible to explore the functional consequences of ubiquitylation with different polyUb chain configurations (the Ub codes) in living cell experiments? (2) Do the functional consequences of Ub code usage depend on the cell type, cellular context, and protein sequences? To achieve their goals, the authors create two sets of molecular tools. First, they create a KCNQ1 construct (BBS-Q1-YFP) that allows fluorescent labeling of cell surface KCNQ1 (fluorescent batrachotoxin that binds to the extracellularly oriented BTX binding sequence fused into KCNQ1), with YFP fused to the cytoplasmic N-terminus that can be used as a read out of total KCNQ1 as well as the tether site of GFP/YFP-nanobody. Second, they fuse the catalytic domains of 4 Ub-linkage preferred DUBs and one nonselective DUB to the GFP/YFP-nanobody (called engineered or enDUBs). The authors validate these molecular tools by

showing that when BBS-Q1-YFP and enDUBs are coexpressed in cells, the degree of Q1 ubiquitylation is drastically reduced. A KCNQ1 construct without YFP or a catalytically dead enDUB does not have this effect.

With these molecular tools, the authors can evaluate the effects of manipulating KCNQ1 ubiquitylation on the total expression, cell surface expression and subcellular distribution of KCNQ1 using high-throughput fluorescence flow cytometry, confocal microscopy, and patch clamp recording. They also use proteomics to explore the pattern of Ub-linkages on KCNQ1 as well as the potential ubiquitylation sites on KCNQ1. These experiments largely answer YES to question (1) listed above.

In terms of question (2), the authors compare the effects of enDUBs on the total and surface expression of KCNQ1 between HEK293 cells and guinea pig ventricular myocytes (cell-type dependence?), between basal conditions vs when KCNQ1 ubiquitylation is markedly elevated by coexpressing KCNQ1 E3 ubiquitin ligase, NEDD4-2 or ITCH (cellular context dependence?), and between WT KCNQ1 vs trafficking-defective KCNQ1 mutants (protein sequence dependence?). These experiments answer YES to question (2).

Overall, the authors provide novel results that are of significance to the field of protein ubiquitylation. Ubiquitylation is a ubiquitous system of protein post-translational modifications, that influence proteins' degradation, trafficking/distribution and thus functions. Ubiquitin (Ub) can be covalently linked to proteins' lysine side chains as monoUb or multi-monoUb, or as straight or branched polyUb chains via linkage through their own 7 Lys side chains as well as the Met1 -NH₂ group. The latter group of straight or branched polyUb chains created limitless configurations of Ub linkage. Our understanding of the complexity of protein ubiquitylation has been greatly aided by the recent development of antibodies, aptamers and affimers that can recognize different Ub-linkages, allowing global pull down followed by proteomics analysis. The current study focuses on decoding Ub-linkages in living cells, that complements the aforementioned Ab(-like)-based/proteomics approaches.

Although this study has many strengths listed above, it would benefit from some additional data and clarifications of some experimental procedures:

1. Immunostaining of organelles: Based on Fig. 1D and Fig. S2, the authors concluded that KCNQ1 is mainly degraded by proteasomes, instead of lysosomes. I would like to suggest that the authors add proteasome immunostaining to Fig. 1A and Fig. 2F-J, along with quantification of KCNQ1-proteasome colocalization.

We thank the reviewer for this suggestion. We have now included proteasome immunostaining and co-localization quantifications to Figs. 1A and 1B, Fig. S1, and Fig. S9. We did not observe a significant difference in proteasome localization of the channels in the presence of enDUBs (Fig. S9), likely due to the transient nature of their residence in this compartment before degradation.

2. Fig. 7 depicts the images of guinea pig ventricular myocytes expressing BBS-Q1-YFP and nano or different enDUBs. The data allowed the authors to quantify total and surface expression of Q1, as well as the expression level of enDUBs. Because ventricular myocytes are 3D structures, the results of image quantification will depend on the z-planes of confocal imaging. Please comment on the following: (a) How did you choose the z-plane of confocal imaging and the reason behind the choice? (b) Why the YFP and 647 images of enDUB-O1/-Cz and /-U21 show such distinct patches? What could be the subcellular compartments of these patches? (c) Have you run 3D imaging of the whole myocyte thickness, and were the results similar to, or different from, the data presented in Fig. 7C and Fig. 7D? (d) It will be of great value to monitor the distribution of secretary- and endocytic-pathway organelles in myocytes. Since you have used green, red, and far-red channels, the blue channel can be used for the organelles.

For our myocyte confocal analysis, we converted the z-slices of the myocyte thickness into 2D maximum intensity projections (MIPs) to estimate YFP and 647 fluorescence intensities from the whole cell. We have now included this detail in the legends and methods section for clarity. We agree with the reviewer that monitoring the distribution of secretary- and endocytic-pathway organelles in myocytes concomitant with channel trafficking would be valuable and we are pursuing such studies with endogenous channels.

3. Fig. 1I and 1J are critical in showing the effectiveness of the catalytic domains of enDUBs. Please clarify the following: (a) What is the nature of the 100 kDa band in both Q1 and Ub IB of the UT (untransfected? not explained in figure legend) lane? (b) If the 100 kDa band in Q1-transfected lanes represents Q1 monomer, why its Ub band intensity is not reduced or even increased in enDUB coexpressed lanes vs the 'nano' lane? (c) Although the Q1 IB image shows distinct monomer, dimer, trimer and tetramer bands, the Ub IB shows a smear. What may cause this smear in Ub IB? Did you quantify Q1 ubiquitylation shown in Fig. 1J based on the distinct Q1 bands or based on the smear of the whole lanes? (d) The Ub IB image shows alarming unevenness in the 'nano' lane, as well as in the following 3 enDUB lanes. This problem calls into question the validity of Fig. 1J. Please consider replacing the Ub IB image with a better 'representative' one.

(a) The 100 KDa band in the untransfected lane was a non-specific band that we only observed in some pulldown trials and not others. A possible explanation for the non-specific band is it represents a dimer of the heavy chain of the antibody used in pulldown experiments. Consistent with this idea, we never observed it in input Western blots. We have replaced the exemplar blot with ones that do not have this non-specific band in the untransfected lane. (b) Since this is an immunoprecipitation blot (vs an input blot), the relative amounts of Q1 pulled down by the antibody from the total lysate may vary and hence is not representative of Q1 expression under these conditions. (c) Since polyubiquitin modifications have varying lengths, they often show

up as smears when separated on a gel. Hence, we quantified the smear of the whole lane for each condition. (d) We have replaced the representative blot shown in Fig 1I as suggested.

4. Fig. 3F: The differences among the BTX-647 decay time courses are not clear, except the cyan curve. Would a semi-log plot be more effective?

Unfortunately, the differences were not better resolved in a semi-log plot. We have reported the half-lives for each enDUB in the results section and figure legend to quantify the differences.

5. Some of the patch clamp experiments were done on coexpressed KCNQ1/KCNE1, while others on KCNQ1 alone. Why this difference? KCNE1 interacts with KCNQ1's C-terminus where the putative Ub sites are located. Does KCNE1 affect KCNQ1 ubiquitylation?

The patch-clamp experiments evaluating the impact of enDUBs on I_{Ks} were all done in the presence of both KCNQ1 and KCNE1. In Fig 4C, we test the effects of the E3 ligases NEDD4-2 and ITCH on KCNQ1 alone. However, we also report the impact of these E3 ligases on I_{Ks} (KCNQ1+KCNE1) in Figs. 5H & 6H.

6. Three minor points: (a) NEDD4-2 or NEDD4L: Both terms were used in the text. Please clarify/unify. (b) Fig. 1E-1G: need info on how the UT negative control was used in quantification of mass spectrometry data. (c) Fig. S1, the Q1-YFP label is misplaced.

(a) Thank you. We now exclusively use NEDD4-2 throughout the manuscript. (b) The untransfected control was used for non-specific background subtraction of measurements from the experimental conditions. We added this detail to the methods section (c) We fixed the label in Fig. S1. Thank you.